# Drought Severity and Frequency Analysis Aided by Spectral and Meteorological Indices in the Kurdistan Region of Iraq

Heman Abdulkhaleq A. Gaznayee [1,*], Ayad M. Fadhil Al-Quraishi [2,*], Karrar Mahdi [3], Joseph P. Messina [4], Sara H. Zaki [1], Hawar Abdulrzaq S. Razvanchy [5], Kawa Hakzi [5], Lorenz Huebner [6], Snoor H. Ababakr [5], Michel Riksen [3] and Coen Ritsema [3]

1 Department of Forestry, College of Agriculture Engineering Science, Salahaddin University, Erbil 44003, Kurdistan Region, Iraq
2 Petroleum and Mining Engineering Department, Faculty of Engineering, Tishk International University, Erbil 44001, Kurdistan Region, Iraq
3 Soil Physics and Land Management Group, Wageningen University and Research, 6700 AA Wageningen, The Netherlands
4 Department of Geography, The University of Alabama, Tuscaloosa, AL 35405, USA
5 Department of Soil and Water, College of Agricultural Engineering Science, Salahaddin University-Erbil, Erbil 44003, Kurdistan Region, Iraq
6 24943 Flensburg, Germany
* Correspondence: heman.ahmed@su.edu.krd (H.A.A.G.); ayad.alquraishi@tiu.edu.iq (A.M.F.A.-Q.)

**Abstract:** In the past two decades, severe drought has been a recurrent problem in Iraq due in part to climate change. Additionally, the catastrophic drop in the discharge of the Tigris and Euphrates rivers and their tributaries has aggravated the drought situation in Iraq, which was formerly one of the most water-rich nations in the Middle East. The Kurdistan Region of Iraq (KRI) also has catastrophic drought conditions. This study analyzed a Landsat time-series dataset from 1998 to 2021 to determine the drought severity status in the KRI. The Modified Soil-Adjusted Vegetation Index (MSAVI2) and Normalized Difference Water Index (NDWI) were used as spectral-based drought indices to evaluate the severity of the drought and study the changes in vegetative cover, water bodies, and precipitation. The Standardized Precipitation Index (SPI) and the Spatial Coefficient of Variation (CV) were used as meteorologically based drought indices. According to this study, the study area had precipitation deficits and severe droughts in 2000, 2008, 2012, and 2021. The MSAVI2 results indicated that the vegetative cover decreased by 36.4%, 39.8%, and 46.3% in 2000, 2008, and 2012, respectively. The SPI's results indicated that the KRI experienced droughts in 1999, 2000, 2008, 2009, 2012, and 2021, while the southeastern part of the KRI was most affected by drought in 2008. In 2012, the KRI's western and southern parts were also considerably affected by drought. Furthermore, Lake Dukan (LD), which lost 63.9% of its surface area in 1999, experienced the most remarkable shrinkage among water bodies. Analysis of the geographic distribution of the CV of annual precipitation indicated that the northeastern parts, which get much more precipitation, had less spatial rainfall variability and more uniform distribution throughout the year than other areas. Moreover, the southwest parts exhibited a higher fluctuation in annual spatial variation. There was a statistically significant positive correlation between MSAVI2, SPI, NDWI, and agricultural yield-based vegetation cover. The results also revealed that low precipitation rates are always associated with declining crop yields and LD shrinkage. These findings may be concluded to provide policymakers in the KRI with a scientific foundation for agricultural preservation and drought mitigation.

**Keywords:** drought; Iraqi Kurdistan Region; normalized difference water index; standardized precipitation index

## 1. Introduction

Drought is a complicated natural disaster that is difficult to diagnose (including its onset, duration, intensity, and scope), forecast, and manage in a broader context; it has

a very negative effect on the social, environmental, and economic status of the affected region [1]. In general, drought results in water scarcity and is caused by low precipitation averages, high evapotranspiration rates, a lack of natural water resources, over-exploitation of water resources, or a combination of these factors [1,2]. Several additional climatic elements have an essential role in the incidence of drought [3], including high temperature, strong winds, relatively low air humidity, timing and rain patterns (particularly during agricultural growth seasons), severity, and length [4,5]. Drought and climate variability, as well as their associated impacts on water resources, have gained increased attention in recent decades as nations seek to enhance mitigation and adaptation mechanisms [2]. Besides precipitation, the most crucial component of the hydrologic budget is water stress, which can result from excessive evapotranspiration rates [6,7], overexploitation of water resources, or a combination of these variables [8]. Drought poses significant hazards to individuals and the environment; hence, it is crucial to understand the spatiotemporal pattern of drought [9]. Various parts of the world are predicted to experience increasingly frequent and severe droughts as a result of climate change [10]. When there is an extended lack of precipitation, meteorologists talk of a meteorological drought [11]. We refer to an agricultural drought when a lack of precipitation results in depleted soil moisture and inadequate plant cover [12].

The periods of drought substantially harmed the agriculture sector and vulnerable populations in the Kurdistan Region [3,13,14]. The Kurdistan Region of Iraq (KRI) has sufficient water resources; however, these supplies are restricted and unpredictable in time and area. According to the Ministry of Agriculture and Water Resources in the KRI, nearly 40% of the KRI's springs dried up during prior droughts. In addition, the water resources in Turkey and Iran [13] mostly depend on the amount of precipitation and seasonal snowfall, as well as the policy of running dams and reservoirs in rivers with shared watersheds. Without international water-sharing agreements between these nations, Iraq's water supplies change from year to year. Water shortage and water quality will be anticipated to deteriorate, especially once Turkey completes its dam projects and Iran builds its planned irrigation projects. In addition, the area anticipates that population expansion, rising water consumption, and climate change will significantly impact water supplies. According to the 2011 Regional Development Strategy for KRI, the Tigris faced a 40% water shortfall in 2016 [15,16].

However, further research is required to comprehend drought events' historical frequency, length, and spatial extent and identify the most susceptible water-using sectors. The studies aid academics, decision-makers, and drought planners in mitigating the negative effects of crisis-based management measures [17–19]. The estimations of surface and groundwater are the primary sources of irrigation water required for agricultural sustainability [20]. Given the limited study on assessing LD in terms of climate change, evaluating how the climate has changed and fluctuated historically in connection to this and other lakes is essential. Moreover, using satellite pictures and remote sensing [21], we investigate the fluctuations in LD's water area extent.

Utilizing remote sensing (RS) techniques for drought monitoring is an efficient and effective method, especially for developing drought indices as well as related spatial data analysis tools, while models and databases also significantly contribute nowadays in predicting, preventing, researching, addressing, rehabilitating, and managing these phenomena of drought [1,22]. This is partly because remote sensing techniques enable more data collection over a larger geographical area and with fewer resources than ground-based observations [22]. Whether the purpose is agricultural, meteorological, or hydrological, satellite data can be exploited for drought monitoring. This data enables one to comprehend the manifestations of drought in a greater region more directly and in less time than previous techniques [23]. Numerous studies utilize meteorological drought indices for drought evaluation, monitoring, and decision-making. The Standardized Precipitation Index (SPI) [24–26] is a frequently employed drought characterization index. This precipitation-based indicator is practical and straightforward. In addition, SPI might be

measured at various intervals during meteorological drought monitoring [27,28]. The Modified Soil-Adjusted Vegetation Index (MSAVI2) is also considered an excellent predictor of dry and semi-arid vegetation cover [29,30], and was designed for low-cover areas to map vegetation in arid mountainous environments [31]. In mountainous areas, primarily topographic gradients govern species distributions; thus, they must be incorporated into the mapping process [32].

The primary objectives of this study are to provide an insight into the historical frequency, duration, and spatial extent of drought episodes and agricultural drought by: (1) analyzing temporal trends in annual total precipitation, vegetation cover, and water body area over the period 1998–2021; (2) calculating the frequency, degree, and variation of drought and drought intensity over the past two decades; and (3) identifying spatial variations in drought and drought rates based on climate variables. Agriculture and water resources in the KRI require such an evaluation and information on vegetation cover to launch vegetation conservation and restoration activities. This study may help decision-makers design better strategies to enhance the KRI's land and water management sector to achieve the second sustainability development goal (SDG 2) adopted by the United Nations (UN) and Nuffic program goals in Iraq for agricultural strategy planners and regional authorities.

## 2. Materials and Methods

### 2.1. Study Area

This study was conducted in the KRI, which is situated between latitudes 33°57′58.5″–37°20′33.55″ N and longitudes 42°20′25.36″–46°19′16.475″ E (Figure 1A), with its elevation ranges between 88 and 3600 m (Figure 1C). It encompassed the governorates of Duhok, Erbil, and Sulaimaniyah. The KRI has a Mediterranean climate that is cold and wet in the winter and hot and dry in the summer [33,34]. Generally, the climate is determined by high precipitation rates in the north and a dryer climate in the plains [35,36]. From October to May, precipitation ranges from 350 mm in the southern regions to more than 1200 mm in the northern and northeastern regions (Figure 1D). The rainfall distribution is unimodal and concentrated from December to April [37]. The average daily temperature ranges from 5 °C in the winter to 30 °C in the summer, but in the south, it can reach 50 °C [37]. Physiographically, the KRI can be divided into the Zagros Mountains and the foothills. The precipitation pattern is influenced by the Mediterranean climate. On the other hand, the KRI is split into three categories based on average annual precipitation: assured rainfall area (above 500 mm), semi-assured rainfall area (350–500 mm), and unassured rainfall area (below 350 mm) [33,34]. Furthermore, the total area of rainfed arable land is 10,682 km$^2$, which accounts for 87.6% of all agricultural land (Figure A1). Approximately 7202 km$^2$ of the KRI's agricultural area is devoted to the production of field crops, constituting a significant share of the KRI's agricultural acreage. Two field crops comprise most of the total land area dedicated to field crops [38].

### 2.2. Datasets

#### 2.2.1. Satellite Images Data

For this study, 144 Landsat images have been downloaded from the Landsat databases on the U.S. Geological Survey website (glovis.usgs.gov (accessed on 28 May 2022)). MSAVI2 was calculated using the Google Earth Engine (GEE). Table A4 displays the JavaScript code used to construct MSAVI2. The images were obtained between 1998 and 2021, during April and May, when yearly vegetation growth was at its highest in the study area. The datasets were gathered from three different Landsat satellites: L5 Thematic Mapper (TM), L7 Enhanced Thematic Mapper Plus (ETM+), and Landsat 8 OLI, which represents the data of (Path/row: 170/34, 170/35, 169/35, 169/34, 168/35, 168/36). Landsat images offer a 30 m spatial resolution (Table A2).

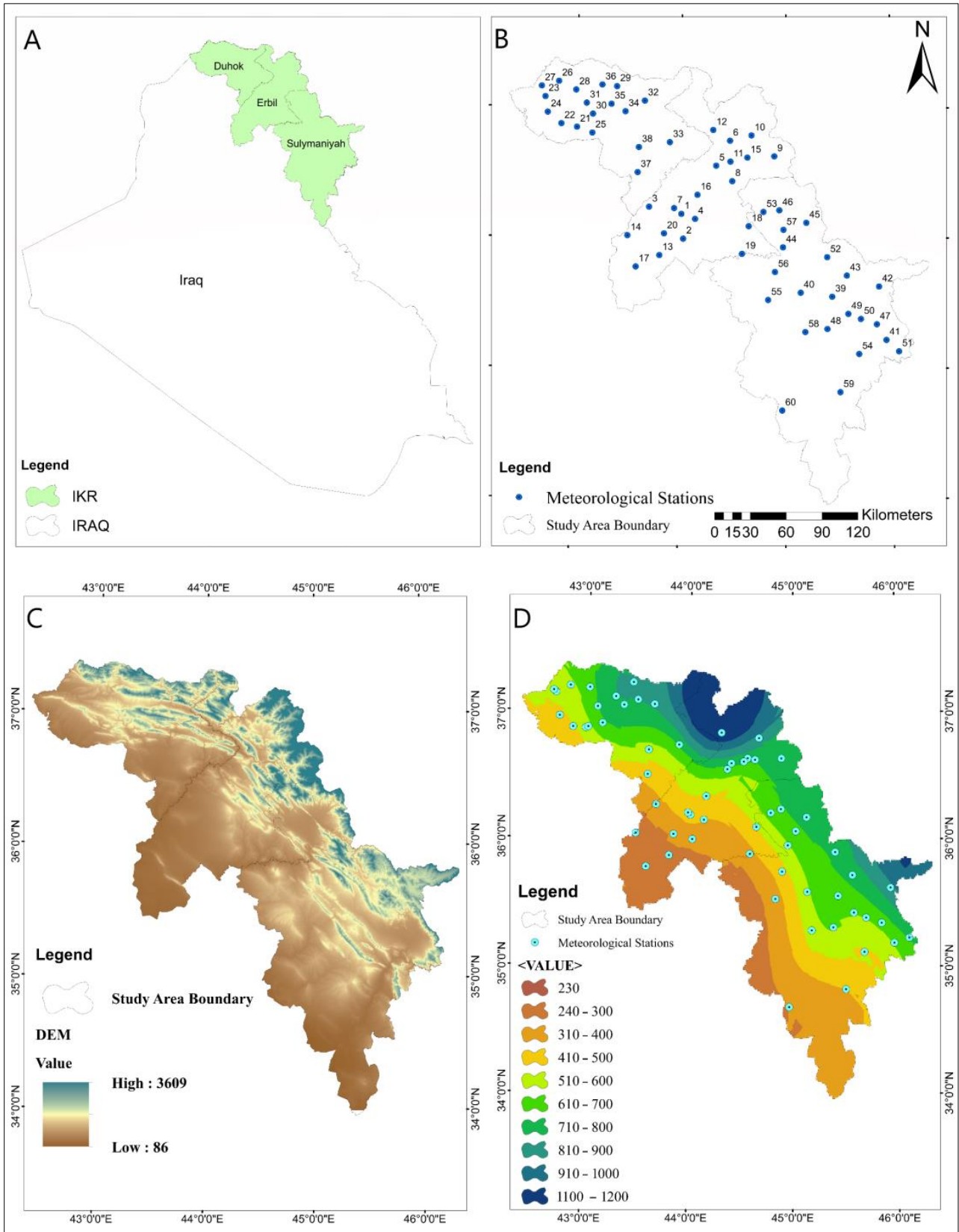

**Figure 1.** (**A**) Site map of the study area in the KRI, (**B**) the weather stations' locations and their codes, (**C**) the elevation map of the study area, and (**D**) the average of 24 years' precipitation (mm) map.

### 2.2.2. Meteorological Data

Data on annual precipitation (AP), and geographical coordinates (longitude, lati-tude, and elevation) for 60 stations were obtained from the Ministry of Agriculture and Water Resources of KRI for the period from 1998 to 2021 (Table A1). Additionally, Figure 1B shows the spatial distribution of these stations. Moreover, Figure 2 shows the overall methodological flowchart utilized in this work, illustrating the whole drought trend analysis procedure. These data were used to estimate the SPI and CV indices.

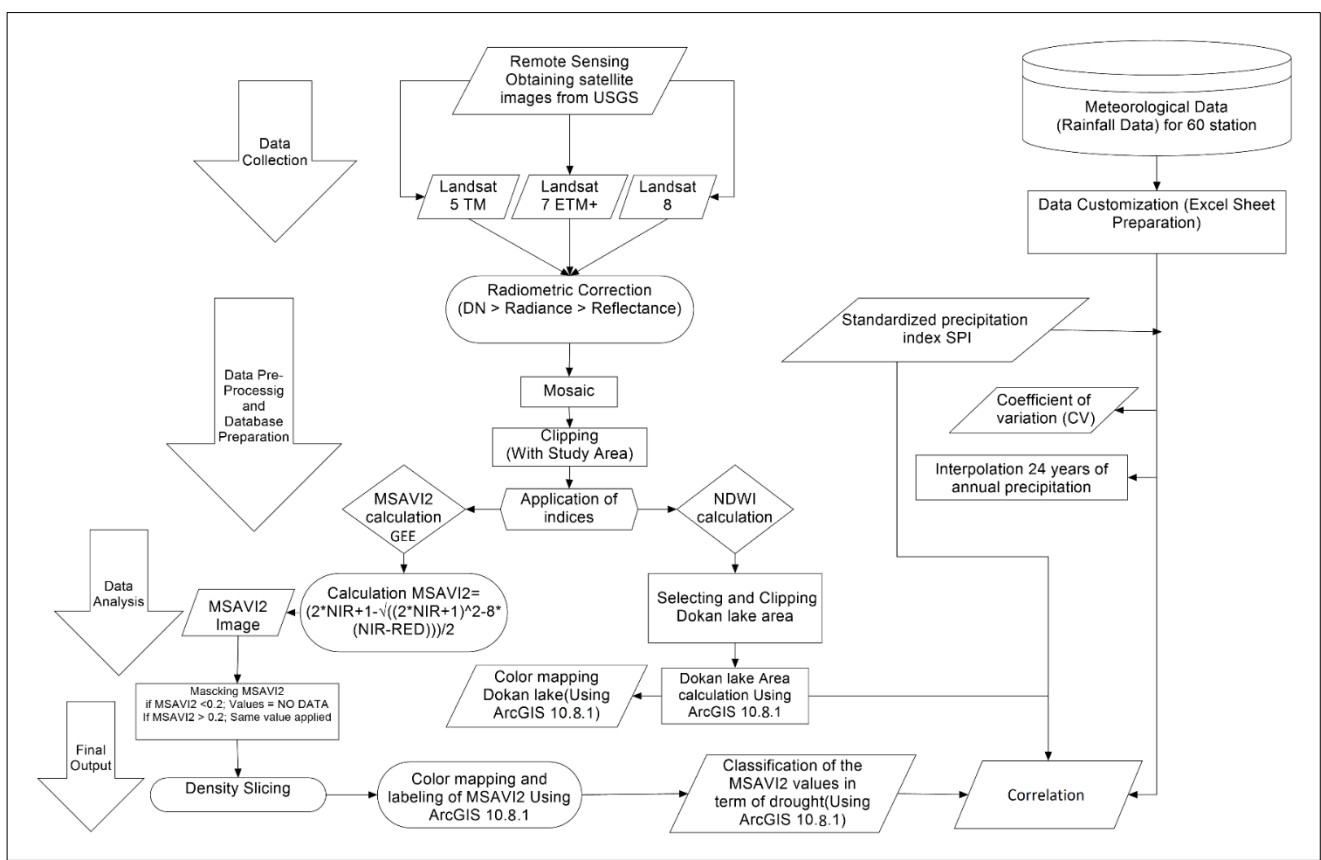

**Figure 2.** Flowchart of the methodology.

### 2.3. Spectral Drought Indices

The spectral datasets were used to calculate the MSAVI2 and NDWI [39] to identify vegetation and drought trends in time and space from a long-term sequence between 1998 and 2021 [40]. Furthermore, using ArcGIS software, 30 m high resolution satellite imagery was used to calculate LD area in km$^2$.

### 2.3.1. The Modified Soil-Adjusted Vegetation Index (MSAVI2)

MSAVI2 is an upgrade to MSAVI; although it is comparable to the SAVI index, it is more accurate for high-exposure soil locations and simply calculates a correction factor for soil brightness [41,42]. MSAVI2 values vary from −1 to +1, with values between −1 and 0 signifying non-plant features such as bare surface, built-up area, and water body, and values greater than 0 representing vegetation cover. The primary objective of this step is to mask non-vegetated areas, such as meadows, residential sites, and roadways, so that only vegetated regions remain. Using the following formula [41,42], MSAVI2 is calculated per pixel:

$$\text{MSAVI2} = \frac{2 * NIR + 1 - \sqrt{(2 * NIR + 1)^2 - 8 * (NIR - RED)}}{2} \tag{1}$$

2.3.2. The Normalized Difference Water Index (NDWI)

LD is located in Sulaimaniyah (SU), between latitudes 35°:30″ and 36°:40″ N and longitudes 44°:30″ and 46°:20″ E. It is considered the largest lake in the KRI and is a reservoir created on the Little Zab River by the Dukan Dam, which was built to provide water storage, irrigation, and hydroelectricity [15,16]. NDWI was employed to map the surface area of LD [15,16,43].

According to McFeeters [44], water bodies can be mapped using a threshold value to separate surfaces with detectable water from those without (NDWI values less than 0.3 vs. NDWI values higher than or equal to 0.3). The NIR and green bands were used to calculate the NDWI according to the equation below.

$$\text{NDWI} = \frac{Green - NIR}{Green + NIR} \tag{2}$$

where *Green* refers to the green wavelengths and *NIR* refers to the near-infrared wavelengths.

*2.4. Meteorological Drought Indices*

2.4.1. Standardized Precipitation Index (SPI)

McKee [24] developed the SPI, which has grown in favor over the past two decades due to it is substantial theoretical development, robustness, and applicability in drought analyses. This study relies on spectral and meteorological indices; therefore, selecting an appropriate index for comparing values across varied climatic regions is crucial. Consequently, the SPI index was used for various analyses [45,46], including frequency and temporal-spatial studies [47]. The SPI is the number of standard deviations from the long-term mean of a normally distributed random variable, which is the observed value in this case [48,49]. The drought severity varied from region to region during the stated drought years. Moreover, the SPI index provides trend analysis for the specified regions. Using DrinC software and the hydrological year (October–September), the default calculation period begins in October with an annual first calculation step. The anomalous strength was categorized after normalized SPI readings, as shown in (Table 1).

**Table 1.** SPI drought severity classes for wet and dry periods [26].

| SPI | Class |
| --- | --- |
| 2.0 or more | Extremely wet |
| 1.5 to 1.99 | Very wet |
| 1.0 to 1.49 | Moderately wet |
| 0.99 to −0.99 | Near normal |
| −1.0 to −1.49 | Moderate drought |
| −1.5 to −1.99 | Severe drought |
| −2.0 or less | Extreme drought |

The SPI is computed by dividing the difference between the normalized seasonal precipitation and its long-term seasonal mean by the standard deviation. It can be calculated using the formula:

$$\text{SPI} = \frac{X_{ij} - X_{im}}{\sigma} \tag{3}$$

where $X_{ij}$ is the seasonal precipitation at the rain gauge station and the observation, $X_{im}$ is the long-term seasonal mean, and $\sigma$ is its standard deviation.

2.4.2. Spatial Distribution of Rainfall across the Study Area

The coefficient of variation (CV) is a statistical measure of the deviation of individual data points from the mean. The higher the CV value, the greater the spatial variability, and vice versa [50]. CV is used to determine the spatial distribution of annual precipitation variability depending on data obtained from 60 locations in the KRI, using ArcGIS and the Kriging spatial interpolation technique. The CV applied to precipitation is especially relevant when comparing the results of two separate surveys or tests with different measures or values. Multiplying the coefficient by 100 is an optional step to calculate a percentage [50]. For example, we compare the results of two tests with varying scoring mechanisms. If sample A has a CV of 12% and sample B has a CV of 25%, then sample B has more variation relative to its mean. The coefficient of variation is expressed as:

$$CV = \frac{\sigma}{\mu} * 100 \tag{4}$$

where: $\sigma$ = standard deviation and $\mu$ = mean.

*2.5. The Statistical Analyses*

The Correlation Coefficient (r)

Bivariate correlations (Pearson correlation coefficient) were adopted in order to find if the variables Crop yield (ton)/year, Crop area (km$^2$), Average SPI (60 stations), LD area (km$^2$), MSAVI2 (Mean Values), and Vegetative cover based on MSAVI2 (km$^2$) are related to one another.

**3. Results**

*3.1. Modified Soil-Adjusted Vegetation Index (MSAVI2)*

The MSAVI2 calculated for the study area from 1998 to 2021 is presented in Table 2 for each year. The lowest mean values of MSAVI2 (0.02, 0.23, and 0.25) were recorded in 2000, 2008, and 2021, respectively. These low values occurred due to the decrease in yearly precipitation, a crucial factor in determining the vegetation cover and MSAVI2 score in those years. The years 2015 and 2016 produced the highest MSAVI2 rating (0.46), indicating greater vegetation cover, as illustrated by Figures 3 and 4. The drought's effects in 2000, 2008, and 2021 suggest that nearly all regions were affected. According to MSAVI2 results, the most substantial loss in vegetation cover in 2000 occurred during the growing season (April and May). Severe drought affected 7865.6 km$^2$ (42.9%), particularly in the KRI's southern, central, and southeastern portions. In 2008, the percentage of land covered by vegetation was 0.2, or 10,018.0 km$^2$. The low vegetation percentage may have resulted from a mismatch between seasonal precipitation and plant needs during the evaluation of the critical growth stage.

Three key factors explained the loss and worsening of the vegetation cover in 2000. Firstly, 1999 was also a drought year, and it may have played a significant role in the return of drought for two consecutive years. Secondly, overgrazing; due to the severe drought in 1999 and 2000, many livestock breeders in central and southern Iraq sought to feed and pasture in the KRI [51,52]. During 1999 and 2000, grasses, bushes, and forests experienced a drastic reduction in vegetation coverage. Thirdly, a physiological explanation is that drought, in most circumstances, results in an incomplete seed production physiological cycle. In addition, it may fail to produce a sufficient number of viable seeds for the bush, pasture, and grass, which substantially impacts the germination of seeds and the growth of vegetation in subsequent years [11,51,53]. Figures 3–5 illustrate the spatial and temporal distribution of MSAVI2 in the KRI from 1998 to 2021. The vegetation cover showed significant spatial variation at the spatial scale, particularly in the middle of the KRI, whereas the northeastern and southern regions remained the most and most minor vegetative areas, respectively. There was an essential relationship between MSAVI2 and precipitation averages across the KRI from 1998 to 2021.

**Table 2.** The max, min, mean, std. dev. of MSAVI2 values and the area of vegetative cover and the MSAVI2—based vegetation density classes in the KRI from 1998 to 2021.

| Years | Max | Min. | Mean | Std. Dev. | Class 1 Values <0.2 Very Low MSAVI2 (km²) | (%) | Class 2 Values 0.2–<0.6 Low to Moderately Low MSAVI2 (km²) | (%) | Class 3 Values 0.6–1 Moderately High to High MSAVI2 (km²) | (%) | Sparse and Non-Vegetation (km²) | Total Vegetative Cover (km²) | (%) | (+ −%) | Total Study Area (km²) |
|---|---|---|---|---|---|---|---|---|---|---|---|---|---|---|---|
| 1998 | 1.00 | 0.20 | 0.42 | 0.15 | 0.0 | 0.0 | 21,347.0 | 86.2 | 3411.3 | 13.7 | 25,506.1 | 24,758.3 | 49.2 | −5.8 | 50,350.6 |
| 1999 | 0.99 | 0.22 | 0.39 | 0.12 | 0.0 | 0.0 | 23,223.8 | 94.6 | 1336.6 | 5.4 | 25,695.5 | 24,560.5 | 48.8 | −6.2 | 50,350.6 |
| 2000 | 0.99 | 0.03 | 0.02 | 0.19 | 7865.6 | 42.9 | 9199.60 | 50.2 | 1274.5 | 6.9 | 31,917.9 | 18,339.6 | 36.4 | −18.5 | 50,350.6 |
| 2001 | 0.84 | 0.19 | 0.41 | 0.14 | 764.9 | 3.3 | 19,843.3 | 86.4 | 2362.8 | 10.2 | 27,289.9 | 22,971.0 | 45.6 | −9.3 | 50,350.6 |
| 2002 | 0.84 | 0.16 | 0.38 | 0.14 | 2906.1 | 10.5 | 22,677.0 | 81.9 | 2111.8 | 7.6 | 22,563.3 | 27,694.9 | 55.0 | 0.0 | 50,350.6 |
| 2003 | 0.84 | 0.13 | 0.38 | 0.15 | 3769.2 | 13.8 | 21,276.2 | 77.7 | 2352.8 | 8.6 | 22,861.1 | 27,398.1 | 54.4 | −0.6 | 50,350.6 |
| 2004 | 0.84 | 0.10 | 0.35 | 0.15 | 5542.4 | 19.2 | 22,003.7 | 76.2 | 1337.6 | 4.6 | 21,371.6 | 28,883.7 | 57.4 | 2.4 | 50,350.6 |
| 2005 | 0.84 | 0.14 | 0.34 | 0.13 | 3813.5 | 15.7 | 19,858.4 | 81.7 | 647.9 | 2.7 | 25,933.4 | 24,319.8 | 48.3 | −6.7 | 50,350.6 |
| 2006 | 0.88 | 0.09 | 0.36 | 0.17 | 5834.1 | 22.6 | 17,734.6 | 68.8 | 2190.5 | 8.5 | 24,499.9 | 25,759.2 | 51.2 | −3.8 | 50,350.6 |
| 2007 | 0.84 | 0.21 | 0.44 | 0.13 | 0.00 | 0.0 | 26,028.6 | 88.5 | 3388.6 | 11.5 | 20,844.9 | 29,417.2 | 58.4 | 3.5 | 50,350.6 |
| 2008 | 0.78 | 0.05 | 0.23 | 0.13 | 10,018 | 50.0 | 9856.50 | 49.2 | 154.60 | 0.8 | 30,222.3 | 20,029.1 | 39.8 | −15.2 | 50,350.6 |
| 2009 | 0.92 | 0.15 | 0.39 | 0.14 | 2348.8 | 9.4 | 20,656.6 | 82.5 | 2030.0 | 8.1 | 25,223.3 | 25,035.4 | 49.7 | −5.2 | 50,350.6 |
| 2010 | 0.84 | 0.23 | 0.43 | 0.12 | 0.00 | 0.0 | 25,131.1 | 89.2 | 3034.1 | 10.7 | 22,096.2 | 28,165.1 | 55.9 | 1.0 | 50,350.6 |
| 2011 | 0.86 | 0.15 | 0.36 | 0.15 | 3540.8 | 14.7 | 18,352.7 | 76.1 | 2217.4 | 9.2 | 26,148.9 | 24,110.9 | 47.9 | −7.1 | 50,350.6 |
| 2012 | 0.84 | 0.10 | 0.35 | 0.15 | 4391.9 | 18.9 | 17,575.1 | 75.5 | 1324.5 | 5.7 | 26,964.8 | 23,291.5 | 46.3 | −8.7 | 50,350.6 |
| 2013 | 0.77 | 0.28 | 0.44 | 0.10 | 0.0 | 0.0 | 26,300.6 | 93.3 | 1880.4 | 6.7 | 22,076.3 | 28,181.0 | 56.0 | 1.0 | 50,350.6 |
| 2014 | 0.77 | 0.30 | 0.45 | 0.09 | 0.0 | 0.0 | 29,578.3 | 93.2 | 2161.0 | 6.8 | 18,518.1 | 31,739.3 | 63.0 | 8.1 | 50,350.6 |
| 2015 | 0.78 | 0.29 | 0.46 | 0.10 | 0.0 | 0.0 | 30,243.0 | 91.6 | 2787.5 | 8.4 | 17,228.6 | 33,030.4 | 65.6 | 10.6 | 50,350.6 |
| 2016 | 0.84 | 0.30 | 0.46 | 0.09 | 0.0 | 0.0 | 29,637.6 | 92.2 | 2498.5 | 7.8 | 18,122.3 | 32,136.1 | 63.8 | 8.9 | 50,350.6 |
| 2017 | 0.78 | 0.30 | 0.44 | 0.09 | 0.0 | 0.0 | 26,111.7 | 96.7 | 896.8 | 3.3 | 23,245.4 | 27,008.5 | 53.6 | −1.3 | 50,350.6 |
| 2018 | 0.90 | 0.20 | 0.30 | 0.28 | 10,529.2 | 32.8 | 15,936.6 | 49.7 | 5593.3 | 17.4 | 18,291.5 | 32,059.1 | 63.7 | 8.7 | 50,350.6 |
| 2019 | 0.93 | 0.20 | 0.36 | 0.16 | 9501.9 | 22.6 | 20,926.6 | 49.9 | 11,547.8 | 27.5 | 8374.3 | 41,976.3 | 83.4 | 28.4 | 50,350.6 |
| 2020 | 0.99 | 0.10 | 0.30 | 0.14 | 10,998.5 | 28.0 | 20,420.2 | 51.9 | 7920.5 | 20.1 | 11,011.4 | 39,339.2 | 78.1 | 23.2 | 50,350.6 |
| 2021 | 0.90 | 0.10 | 0.25 | 0.12 | 10,772.9 | 44.9 | 10,707.3 | 44.6 | 2535.9 | 10.6 | 26,334.5 | 24,016.1 | 47.7 | −7.3 | 50,350.6 |

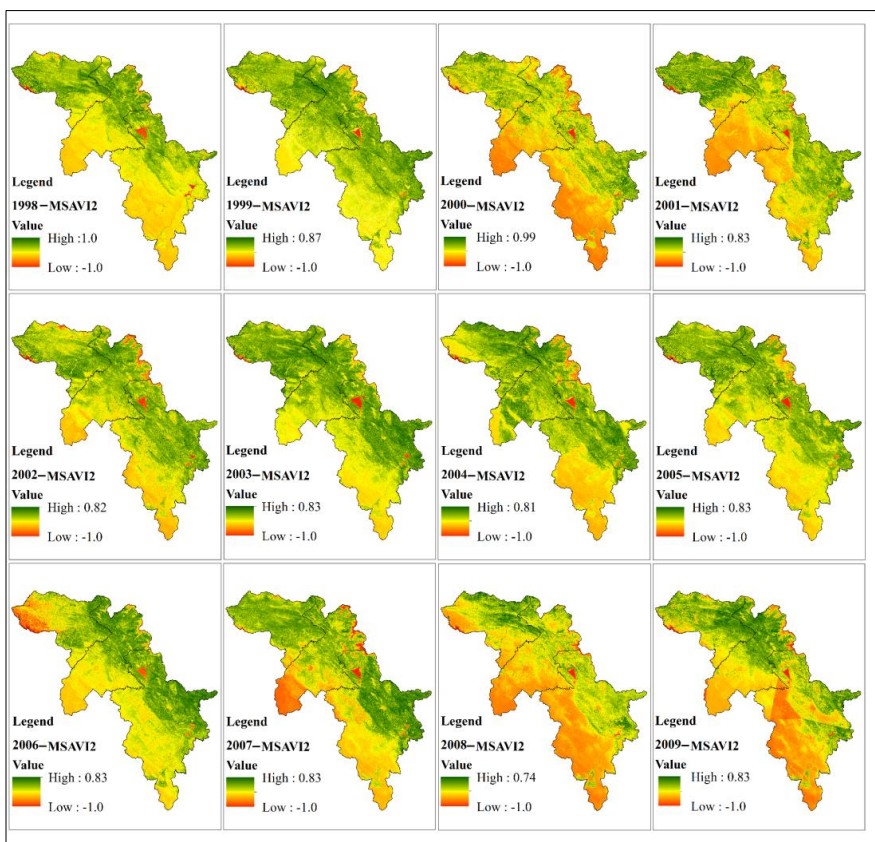

**Figure 3.** Spatial variation of the MSAVI2−based vegetation from 1998 to 2009.

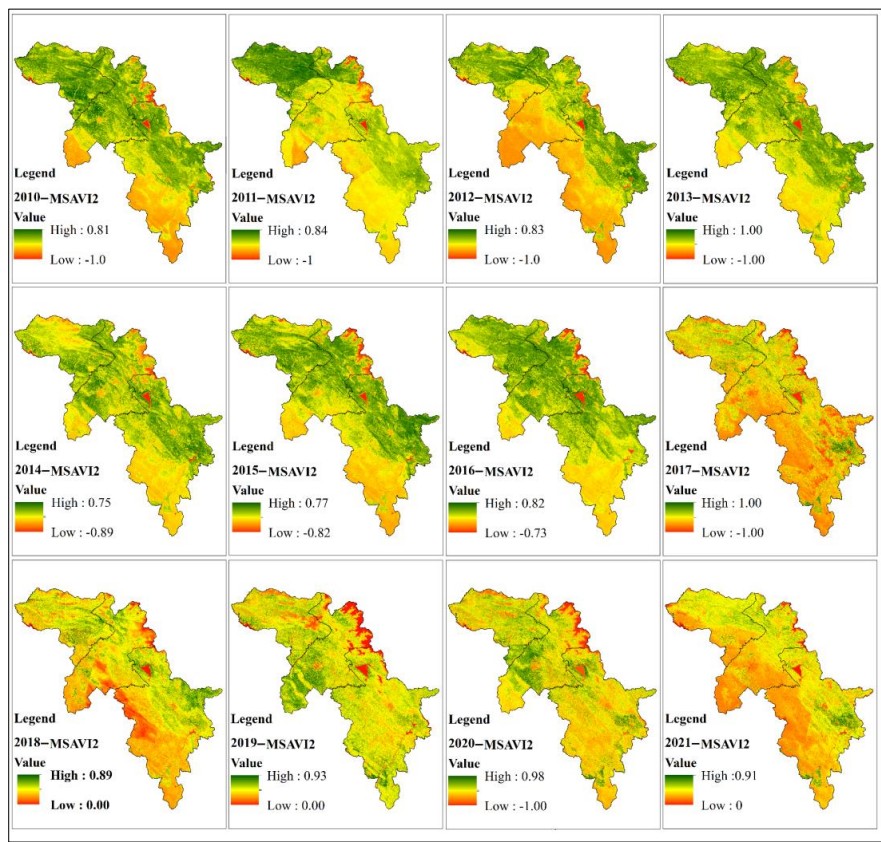

**Figure 4.** Spatial variation of the MSAVI2−based vegetation from 2010 to 2021.

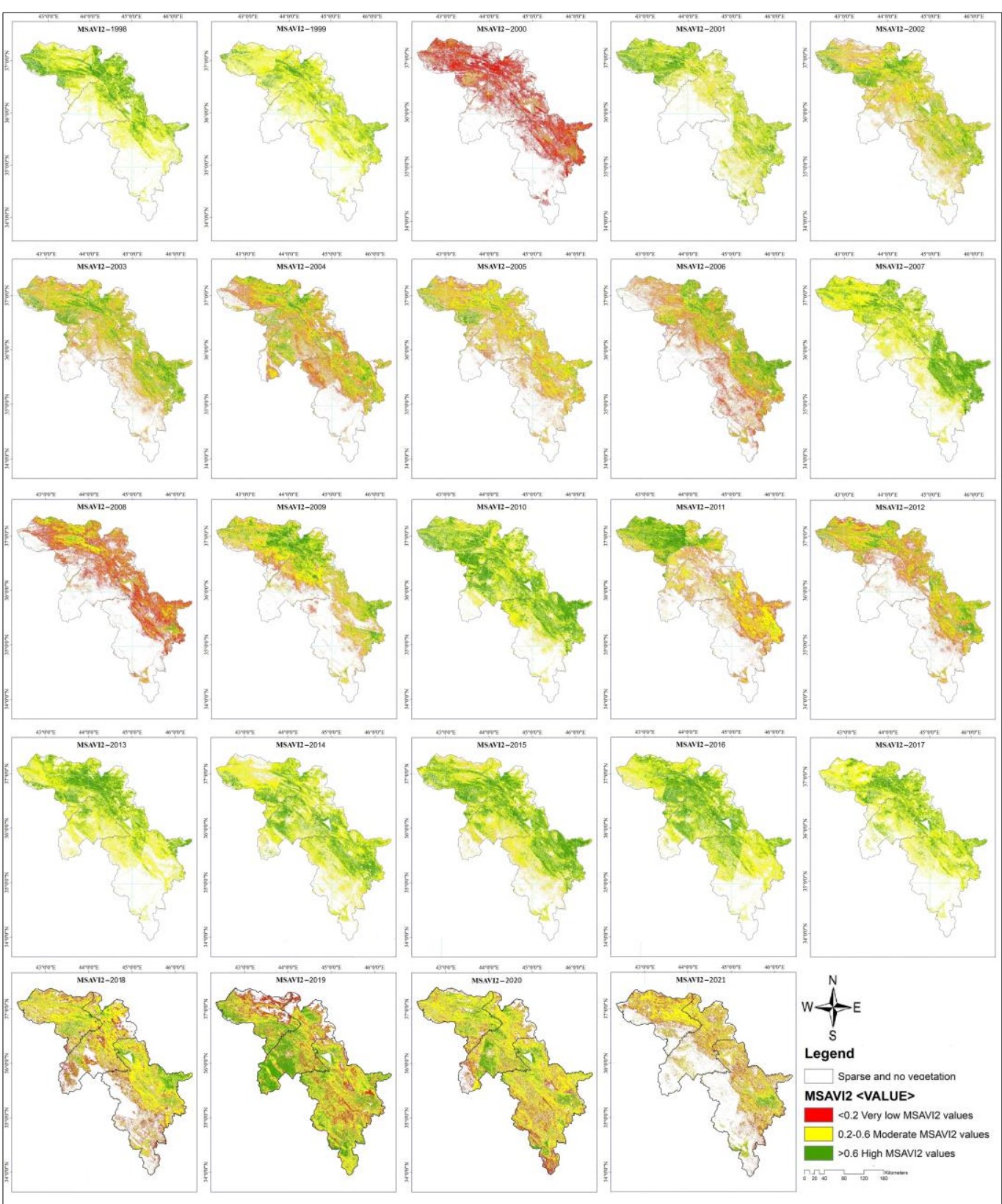

**Figure 5.** Spatial variation of the MSAVI2−based vegetation density classes from 1998 to 2021.

This trend is consistent with the meteorological features of the study area, namely the average rainfall and temperature. In general, precipitation was highest (about 1000 mm) in the northeast and gradually decreased in the southwest (to around 150 mm). Additionally,

elevation followed the same pattern of decline and indirectly influenced temperature and precipitation. The association between vegetation cover area and MSAVI2 values and elevation data was statistically significant [14]. The results reported in Table 2 and Figures 3–5 make this very obvious. Based on the results presented in Table 2 and Figures 3–5, the years 2000, 2008, 2012, and 2021 were the most vulnerable to drought, as detected by the vegetation growth in the region.

In comparison to earlier years, the vegetation cover was drastically reduced throughout these years. During the most severe drought in 2000, the vegetative cover was reduced to 18,339.6 km$^2$ (representing 36.4% of the overall study area). During 1998–2021, the average vegetation coverage was 55%, although the vegetation coverage in 2000 varied by 36.4% from the average, and the vegetative cover area declined to 24016.1 km$^2$ in 2021 (representing 47.7% of the overall study area).

### 3.2. NDWI (Waterbody Area of LD)

The spatiotemporal analysis found that LD reached its greatest extent of 282 km$^2$ in 2019 and its smallest extent of 125 km$^2$ in 2009 (Table 3 and Figures 6 and 7). In addition, Figures 6 and 7 and Table 3 show that the most severe droughts in the LD area occurred during the hydrologic years 1999, 2000, 2008, and 2009, by 140 km$^2$, 137 km$^2$, 135 km$^2$, and 125 km$^2$, respectively. Numerous causes, such as bordering countries prohibiting water imports and territorial laws, decreasing yearly precipitation, constructing various dams in all riparian countries, and rising water demand for agricultural activities, have been attributed to the LD level decline [13]. Low water levels have resulted from drought years in Iraq's river basins, particularly the Tigris, which contributes 70% of the country's water resources [54].

**Table 3.** Area of water body in (LD) for 1998–2021.

| Time, Year | (LD) Area (km$^2$) | Area Ave. | % (+ −) |
|:---:|:---:|:---:|:---:|
| 1998 | 258 | 195 | 62 |
| 1999 | 140 | 195 | −55 |
| 2000 | 137 | 195 | −58 |
| 2001 | 185 | 195 | −10 |
| 2002 | 225 | 195 | 30 |
| 2003 | 267 | 195 | 72 |
| 2004 | 254 | 195 | 59 |
| 2005 | 238 | 195 | 43 |
| 2006 | 216 | 195 | 21 |
| 2007 | 189 | 195 | −6 |
| 2008 | 135 | 195 | −60 |
| 2009 | 125 | 195 | −70 |
| 2010 | 159 | 195 | −37 |
| 2011 | 137 | 195 | −59 |
| 2012 | 170 | 195 | −26 |
| 2013 | 200 | 195 | 5 |
| 2014 | 158 | 195 | −37 |
| 2015 | 149 | 195 | −46 |
| 2016 | 229 | 195 | 33 |
| 2017 | 224 | 195 | 28 |
| 2018 | 207 | 195 | 12 |
| 2019 | 282 | 195 | 87 |
| 2020 | 220 | 195 | 25 |
| 2021 | 185 | 195 | −10 |

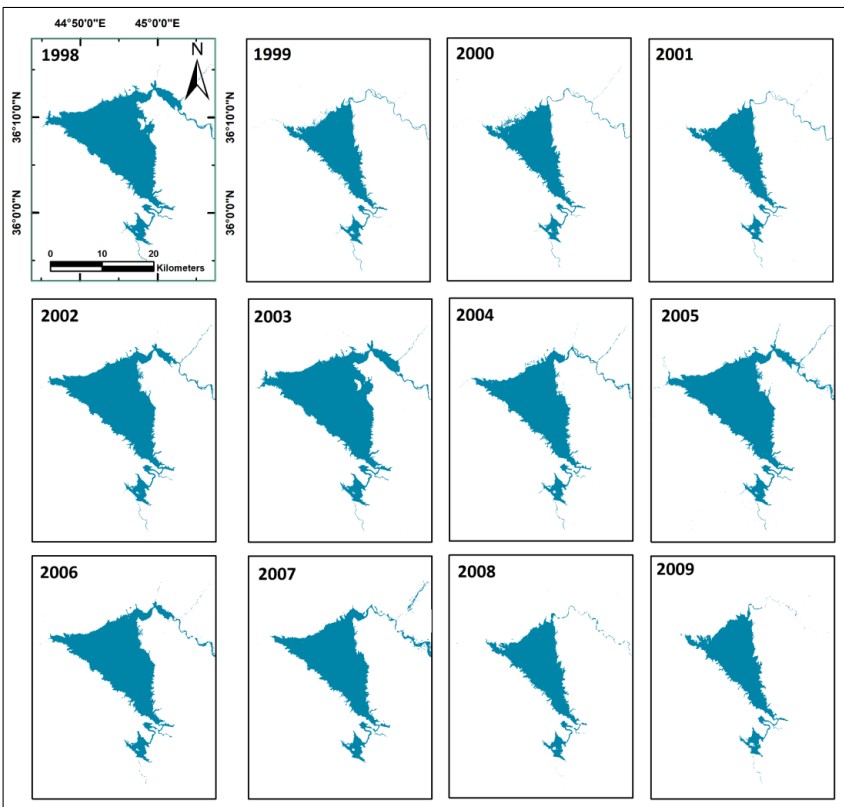

**Figure 6.** Dukan Lake area change from 1998–2009.

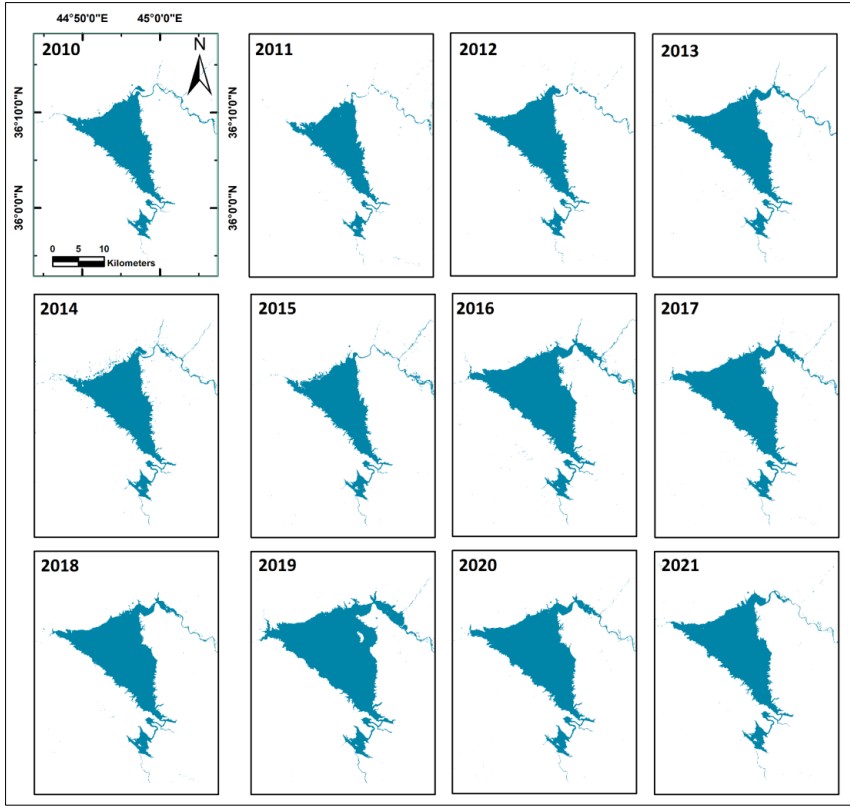

**Figure 7.** Dukan Lake area change from 2010–2021.

From 1997 to 2002, the average annual discharge of the Tigris River into Iraq fell below 43,000 million cubic meters, and from 1997 to 2001, it dropped precipitously to less than 19,000 million cubic meters, or about 40% less than the average annual discharge. Some of the low discharges are attributed to decreased precipitation in the Tigris River watersheds, which is consistent with the expected drop in precipitation in the country due to climate change [55–57]. According to [58], precipitation in the Turkish highlands is anticipated to decline by 10–60% by the end of the century, resulting in a 29% reduction in Tigris flow. According to a study undertaken at the University of California, Irvine, the total water storage in the Tigris and Euphrates rivers, which flow through Turkey, Syria, Iraq, and Iran, is diminishing at an alarming rate. Between 2003 and 2009, the researchers discovered that the river basin lost around 144 km$^3$ of fresh water [16]. Approximately 60% of this loss is related to groundwater extraction from aquifers, which is frequently used to meet demand when surface water resources are insufficient [36].

### 3.3. Standardized Precipitation Index SPI

Figure 8 depicts the spatiotemporal trends of SPI for 60 meteorological stations in the KRI. During the drought years, the severity varied from area to area. According to McKee et al. [26], drought arises when the SPI value is negative and disappears when the SPI value is positive. Four years in the studied historical record, specifically 1999, 2000, 2008, and 2021, saw severe drought, measured by the SPI values. Two years, 2009 and 2012, experienced moderate drought (Figures 8 and 9). Stations 11, 16, 19, 22, 23, 24, 40, 48, 49, 55, 56, 57, and 58 had the most severe drought in 2008, with average SPI values of −2.28, −2.26, −2.27, −2.25, −2.19, −2.54, −2.38, −2.92, −2.24, −2.17, −2.56, −2.35, and −2.28, respectively (Table A3).

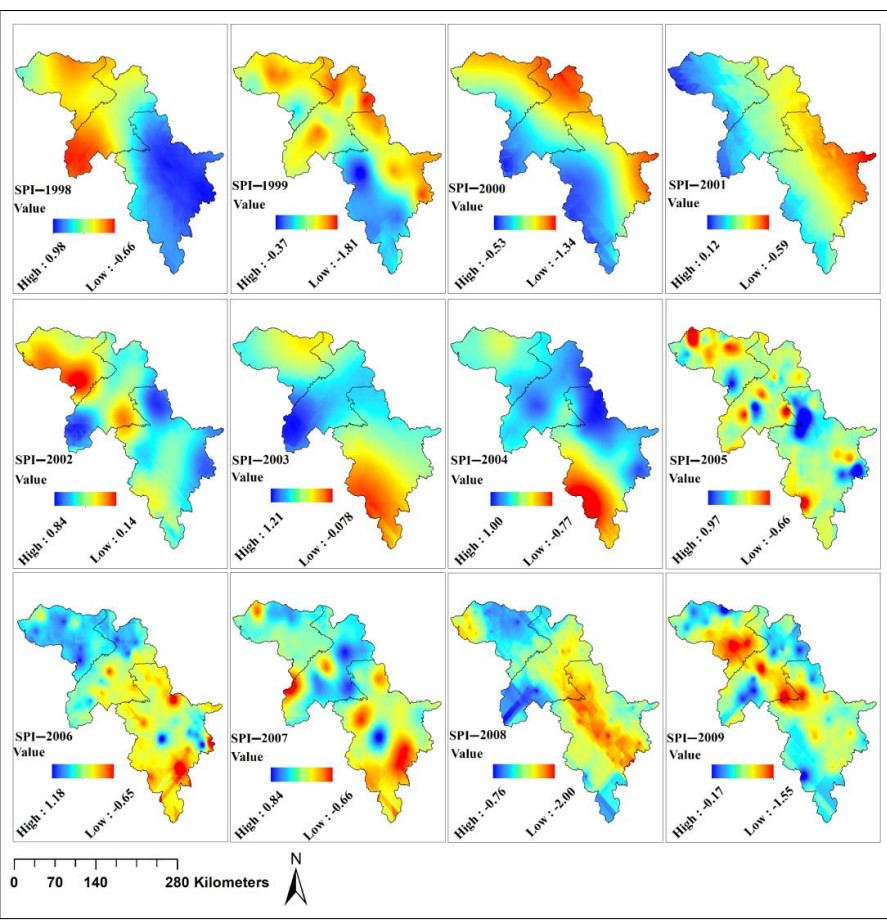

**Figure 8.** Temporal pattern of SPI drought and wet periods for 60 meteorological stations from 1998 to 2009.

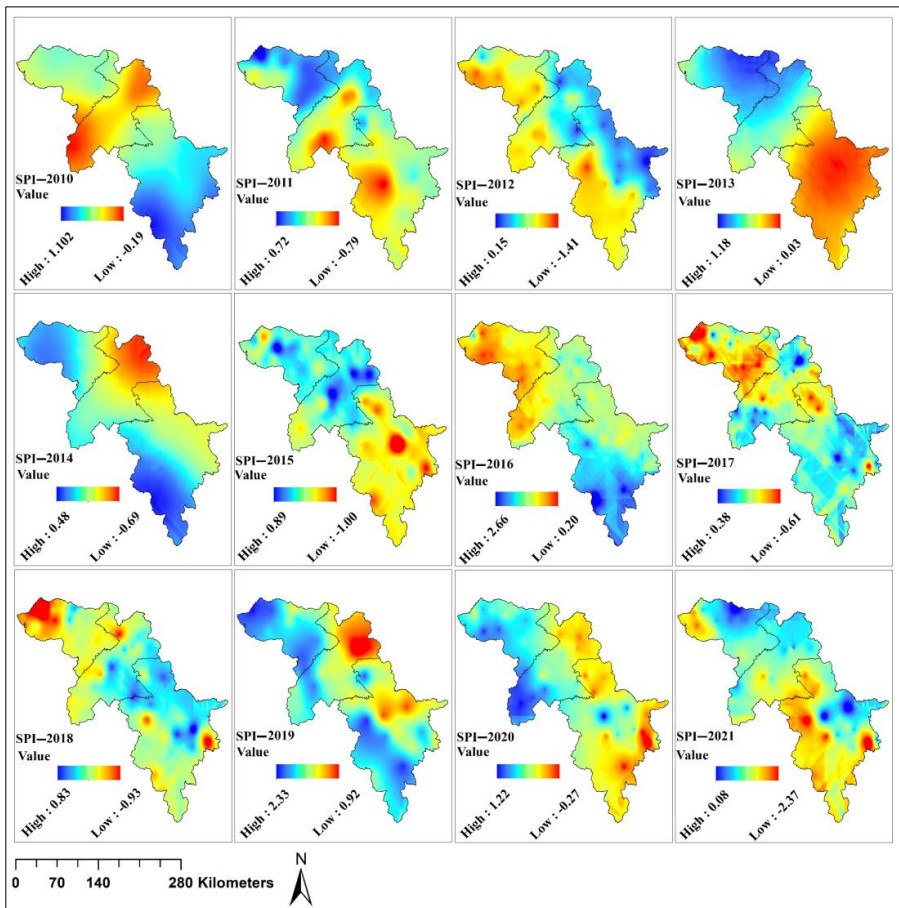

**Figure 9.** Temporal pattern of SPI drought and wet periods for 60 meteorological stations from 2009 to 2021.

In addition, approximately 60% of the studied years fell under the near-normal drought class. The range of the normalized precipitation index for the near-normal class is between −1.0 and 1.0. Overall, there is no discernible trend in the SPI values across the study years (Table 4), with negative and positive SPI values fluctuating over the study period (Figures 8 and 9). However, a comprehensive review of this graph revealed three instances of more severe drought, notably 1999, 2000, 2008, 2012, and 2021 (Figures 8 and 9). The precipitation deficits continued for at least three years, making the drought throughout these three eras long-term. Indeed, dry years were marked by poor river flow, low groundwater and reservoir levels, extremely dry soil, and decreased crop yields or crop failure [59]. Regardless of the severity of the drought, the entire study area in 2008 and 2012 suffered exceptional dryness.

The SPI values computed for each site revealed that the frequency of severe drought to extreme drought has risen in the KRI by more than three to four times during the previous 24 years. This study demonstrates that severe and intense drought occurred intermittently over the study area, resulting in varying implications on agricultural practices and water supplies in the KRI. The spatiotemporal patterns of SPI distribution for 60 meteorological stations in the KRI's sub-districts indicated drought had varying severity in most studied areas between 1999 and 2021. The severity of these drought years varied from area to area. Figures 8 and 9 demonstrate the trend of drought severity for each of the 60 KRI stations.

**Table 4.** The frequency of drought SPI index in 60 weather stations during the 24 years.

| SPI Class | | Extremely Wet | Very Wet | Moderately Wet | Near Normal | Moderate Drought | Severe Drought | Extreme Drought |
|---|---|---|---|---|---|---|---|---|
| Station No. | Station Name | 2.00 or More | 1.50 to 1.99 | 1.00 to 1.49 | 0.99 to −0.99 | −1.00 to −1.49 | −1.50 to −1.99 | −2 or Less |
| Erbil | | | | | | | | |
| 1 | Erbil | 0 | 2 | 1 | 16 | 2 | 1 | 2 |
| 2 | Qushtapa | 0 | 1 | 1 | 18 | 1 | 1 | 2 |
| 3 | Khabat | 1 | 0 | 3 | 15 | 3 | 1 | 1 |
| 4 | Bnaslawa | 0 | 1 | 3 | 17 | 1 | 1 | 1 |
| 5 | Harir | 0 | 1 | 4 | 16 | 1 | 2 | 0 |
| 6 | Soran | 0 | 0 | 7 | 14 | 1 | 2 | 0 |
| 7 | Shaqlawa | 0 | 2 | 1 | 17 | 2 | 2 | 0 |
| 8 | Khalifan | 0 | 1 | 4 | 16 | 1 | 0 | 2 |
| 9 | Choman | 0 | 1 | 3 | 17 | 1 | 1 | 1 |
| 10 | Sidakan | 0 | 1 | 3 | 16 | 1 | 2 | 1 |
| 11 | Rwanduz | 0 | 0 | 6 | 13 | 4 | 0 | 1 |
| 12 | Mergasur | 0 | 1 | 3 | 17 | 1 | 0 | 2 |
| 13 | Dibaga | 1 | 2 | 4 | 12 | 3 | 2 | 0 |
| 14 | Gwer | 1 | 2 | 1 | 14 | 5 | 1 | 0 |
| 15 | Barzewa | 1 | 0 | 0 | 20 | 2 | 1 | 0 |
| 16 | Bastora | 0 | 1 | 3 | 18 | 0 | 0 | 2 |
| 17 | Makhmor | 0 | 2 | 3 | 15 | 3 | 0 | 1 |
| 18 | Koya | 0 | 2 | 2 | 17 | 1 | 1 | 1 |
| 19 | Taqtaq | 0 | 2 | 1 | 16 | 3 | 0 | 2 |
| 20 | Shamamk | 2 | 0 | 3 | 15 | 2 | 1 | 1 |
| *Duhok* | | | | | | | | |
| 21 | Duhok | 2 | 2 | 9 | 7 | 4 | 0 | 0 |
| 22 | Semel | 1 | 1 | 13 | 6 | 1 | 2 | 0 |
| 23 | Zakho | 2 | 1 | 11 | 7 | 1 | 2 | 0 |
| 24 | Batel | 1 | 3 | 9 | 8 | 2 | 1 | 0 |
| 25 | Dam-DU | 2 | 1 | 9 | 9 | 3 | 0 | 0 |
| 26 | Darkar.H | 1 | 4 | 7 | 10 | 2 | 0 | 0 |
| 27 | Zaxo-A.S | 2 | 0 | 12 | 7 | 2 | 1 | 0 |
| 28 | Batifa | 1 | 2 | 11 | 8 | 0 | 2 | 0 |
| 29 | Kani Masi | 1 | 2 | 10 | 8 | 3 | 0 | 0 |
| 30 | Zaweta | 2 | 2 | 10 | 7 | 2 | 1 | 0 |
| 31 | Mangish | 1 | 3 | 9 | 10 | 0 | 1 | 0 |
| 32 | Deraluke | 0 | 4 | 8 | 10 | 0 | 2 | 0 |
| 33 | Akre | 1 | 3 | 10 | 7 | 3 | 0 | 0 |
| 34 | Amadia | 1 | 3 | 8 | 11 | 0 | 1 | 0 |
| 35 | Sarsink | 1 | 2 | 12 | 8 | 0 | 1 | 0 |
| 36 | Bamarni | 0 | 5 | 8 | 9 | 2 | 0 | 0 |
| 37 | Bardarash | 2 | 3 | 7 | 8 | 4 | 0 | 0 |
| 38 | Qasrok | 1 | 2 | 11 | 8 | 2 | 0 | 0 |
| *Sulaimaniyah* | | | | | | | | |
| 39 | **SU** | 0 | 2 | 3 | 15 | 3 | 0 | 1 |
| 40 | Bazian | 0 | 0 | 5 | 16 | 1 | 1 | 1 |
| 41 | Halabja | 0 | 1 | 4 | 15 | 1 | 2 | 1 |
| 42 | Penjwen | 0 | 1 | 2 | 18 | 1 | 0 | 2 |
| 43 | Chwarta | 0 | 0 | 6 | 14 | 2 | 2 | 0 |
| 44 | Dukan | 0 | 2 | 3 | 15 | 2 | 1 | 1 |
| 45 | Qaladiza | 0 | 2 | 3 | 16 | 0 | 2 | 1 |
| 46 | Rania | 0 | 1 | 4 | 15 | 2 | 2 | 0 |
| 47 | Said Sadiq | 1 | 2 | 1 | 15 | 4 | 1 | 0 |
| 48 | Qaradagh | 0 | 2 | 0 | 18 | 3 | 0 | 1 |
| 49 | Arbat | 1 | 1 | 3 | 16 | 1 | 1 | 1 |
| 50 | K-Panka | 0 | 1 | 4 | 15 | 2 | 2 | 0 |
| 51 | Byara | 0 | 1 | 3 | 17 | 1 | 2 | 0 |
| 52 | Mawat | 0 | 2 | 2 | 15 | 3 | 1 | 1 |
| 53 | Dar-Dikhan | 1 | 1 | 3 | 14 | 3 | 2 | 0 |
| 54 | Chamchamal | 0 | 2 | 2 | 15 | 3 | 1 | 1 |
| 55 | Kalar | 2 | 1 | 2 | 17 | 0 | 1 | 1 |
| 56 | Agjalar | 0 | 1 | 4 | 16 | 3 | 0 | 0 |
| 57 | Bngrd | 0 | 1 | 4 | 14 | 3 | 1 | 1 |
| 58 | Sangaw | 1 | 0 | 4 | 15 | 2 | 1 | 1 |
| 59 | Bawanor | 2 | 0 | 1 | 17 | 3 | 0 | 1 |
| 60 | Kifri | 1 | 1 | 2 | 17 | 3 | 0 | 0 |

Figures 8 and 9 show that the drought zone was determined by interpolating SPI data using Kriging. The KRI's SPI values from 1998 through 2021 are displayed in Tables 4 and A3. The data suggest an irregular cyclical pattern of dry/wet spells during the past 24 years. The initial decline in SPI values began in 1999 and continued until 2001. This drop closely parallels the precipitation decrease seen in DU, ER, and SU provinces during the same year. The SPI index findings were determined to be comparable to the NDWI index results (Table 3). In 1999 and 2000, the drought was extremely severe, but in 2008, the part of the KRI worst hit was the southeast. In 2008 and 2012, the western and southern portions of the study area suffered moderate drought. Figures 8 and 9 depicted the SPI values when drought conditions were found in 1999, 2000, 2008, 2012, and 2021.

In comparison, the wettest years were 2003, 2016, and 2019, respectively. According to McKee et al. (1993), drought occurs when the SPI value is negative and dissipates when it is positive. Table 4 demonstrates that around 57% of the studied years fell into the near-normal drought class, with an SPI range of $-1.0$ to $1.0$ for the near-normal class. Table 2 displays that negative and positive SPI values alternate over the study period. The SPI values show no clear trend throughout the studied periods (Tables 4 and A3). However, a closer look (Figures 8 and 9) revealed that the drought was more severe in 1999, 2000, and 2008 than in any other studied year. Drought can occur despite average precipitation in hydrological and vegetative realms [26]. During the growing season, the absence of a relationship between vegetative and each hydrological drought and SPI is most evident.

### 3.4. Spatial Pattern Variation of Precipitation

The Zagros Mountains receive the most precipitation from October through May. To examine the spatial pattern of precipitation variability over the study area, which encompasses the whole KRI, the CV was calculated for each of the 60 study stations. Figure 10 and Table 5 depict the average ($24-$year) precipitation (mm), maximum precipitation (mm), lowest precipitation (mm), standard deviation (%), and coefficient of variation (%). The annual precipitation variability indicates that station #60, with a CV of 56.7%, displayed the most temporal variability, while station #10 exhibited the least, with a CV of 23.0%. Similarly, stations #10 and #17 had the highest and lowest annual precipitation averages, with 1370.3 mm and 244.3 mm, respectively (Table 5).

Generally, the highest CV values are seen in the study area's southern parts, which receive the least precipitation. The statistical results in Table 5 reveal that annual precipitation varies significantly over time. The CV ranged from a low of around 23.1% at station #10 to a high of approximately 56.7% at station #60. (Figure 10). In addition, the lowest annual precipitation averages, less than 244 mm, 297.4 mm, and 293.1 mm at stations #17, #20, and #60, respectively, occurred in low-latitude and low-elevation portions of the KRI. In contrast, higher than 1370.3 mm of precipitation was reported at station #12 in the northern area of the KRI. As the temperature falls with increasing height, Figure 8 depicts a rise from all directions toward high-elevation parts. The spatial variation study reveals, in Figure 10, that the northeast area, which received much more precipitation than other parts, had less regional rainfall variability and a more uniform rainfall distribution than other parts.

In the southwest area, station numbers 2, 13, 14, 15, and 20 (ER), 29, 31,32, 35, and 38 (DU), and 58, 59, and 60 (SU) exhibited a large range of annual spatial variation, with CVs of 43.9, 46.7, 51.5, and 42.8%, and 44.0, 44.3, and 41.5%, respectively. The CV was utilized for the analysis of variability. The study findings showed a downward trend in the KRI's annual and seasonal rainfall series. At station #12 (1370.3 mm) and station #17 (244.3 mm), the maximum and minimum annual precipitation averages, respectively, were recorded. In the southern parts of the KRI, the CV% exhibited significant interannual fluctuation. Figure A2 in Appendix A depicts the CV annual precipitation at 60 selected meteorological stations throughout the KRI. The highest CV values are recorded in the southern parts, characterized by low rainfall.

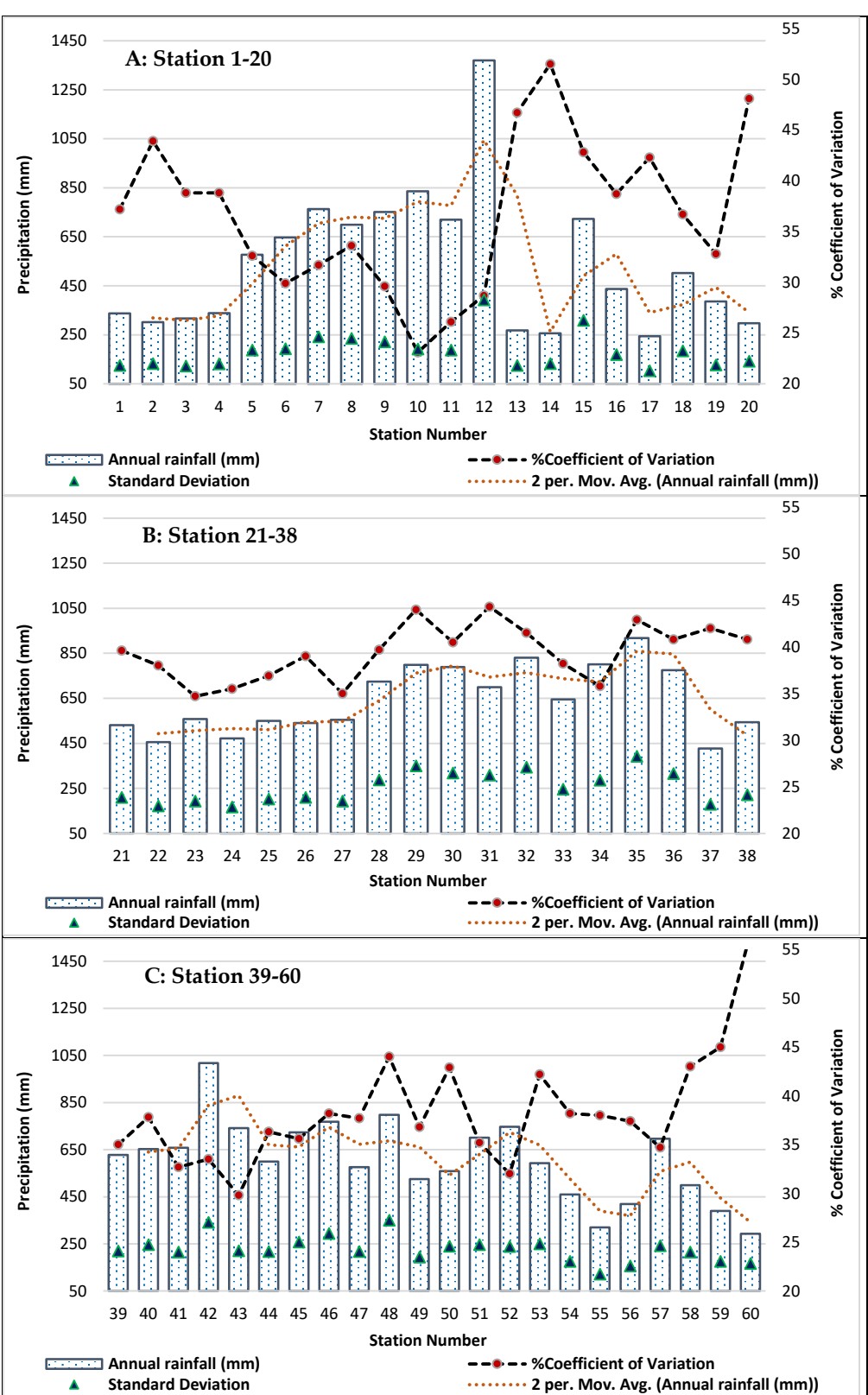

**Figure 10.** Descriptive statistics of the average annual precipitation series data recorded at each of the 60 weather stations.

**Table 5.** Descriptive statistics of the annual precipitation series data recorded at each of the 60 weather stations.

| Station No. | Geographical Coordinates | | Record (Years) | Maximum Rainfall (mm) | Minimum Rainfall (mm) | Average (Annual Rainfall) (mm) | Standard Deviation | Coefficient of Variation CV |
|---|---|---|---|---|---|---|---|---|
| | Longit | Latitude | | | | | | |
| | | | | *Erbil* | | | | |
| 1 | 44.009 | 36.191 | 24 | 645.6 | 114.2 | 337.3 | 125.4 | 37.2 |
| 2 | 44.028 | 36.001 | 24 | 681.5 | 106.1 | 301.3 | 132.2 | 43.9 |
| 3 | 43.674 | 36.273 | 24 | 733.0 | 125.7 | 317.0 | 122.9 | 38.8 |
| 4 | 44.140 | 36.154 | 24 | 694.1 | 118.0 | 338.9 | 131.6 | 38.8 |
| 5 | 44.365 | 36.551 | 24 | 1042.1 | 264.5 | 576.8 | 188.0 | 32.6 |
| 6 | 44.561 | 36.638 | 24 | 963.3 | 290.5 | 647.2 | 193.6 | 29.9 |
| 7 | 43.985 | 36.209 | 24 | 1295.5 | 360.5 | 762.9 | 241.9 | 31.7 |
| 8 | 44.404 | 36.599 | 24 | 1241.3 | 263.6 | 699.3 | 235.3 | 33.6 |
| 9 | 44.889 | 36.637 | 24 | 1131.0 | 271.3 | 750.8 | 221.9 | 29.6 |
| 10 | 44.671 | 36.797 | 24 | 1173.0 | 463.7 | 835.3 | 192.6 | 23.1 |
| 11 | 44.525 | 36.612 | 24 | 1012.4 | 342.4 | 719.6 | 188.0 | 26.1 |
| 12 | 44.306 | 36.838 | 24 | 2111.1 | 624.7 | 1370.3 | 392.9 | 28.7 |
| 13 | 43.805 | 35.873 | 24 | 663.9 | 94.0 | 267.5 | 125.0 | 46.7 |
| 14 | 43.481 | 36.045 | 24 | 601.6 | 93.0 | 256.6 | 132.2 | 51.5 |
| 15 | 44.633 | 36.627 | 24 | 1889.0 | 284.2 | 722.9 | 309.3 | 42.8 |
| 16 | 44.160 | 36.339 | 24 | 870.4 | 139.7 | 436.8 | 169.1 | 38.7 |
| 17 | 43.583 | 35.783 | 24 | 530.3 | 92.0 | 244.3 | 103.4 | 42.3 |
| 18 | 44.648 | 36.099 | 24 | 1047.6 | 216.8 | 501.8 | 184.1 | 36.7 |
| 19 | 44.586 | 35.887 | 24 | 677.6 | 154.9 | 386.2 | 126.7 | 32.8 |
| 20 | 43.847 | 36.040 | 24 | 746.4 | 91.0 | 297.4 | 142.9 | 48.1 |
| | | | | *Duhok* | | | | |
| 21 | 42.979 | 36.868 | 24 | 1120.0 | 217.2 | 531.0 | 210.0 | 39.6 |
| 22 | 42.854 | 36.873 | 24 | 995.0 | 142.7 | 455.5 | 172.9 | 38.0 |
| 23 | 42.682 | 37.144 | 24 | 1165.4 | 232.5 | 557.9 | 193.8 | 34.7 |
| 24 | 42.722 | 36.959 | 24 | 1004.0 | 157.4 | 472.2 | 167.4 | 35.5 |
| 25 | 43.003 | 36.876 | 24 | 1135.0 | 233.1 | 550.0 | 202.9 | 36.9 |
| 26 | 42.823 | 37.199 | 24 | 1187.0 | 242.0 | 540.4 | 210.7 | 39.0 |
| 27 | 42.659 | 37.160 | 24 | 1165.4 | 247.8 | 554.0 | 194.1 | 35.0 |
| 28 | 43.013 | 37.184 | 24 | 1705.5 | 257.2 | 724.8 | 288.1 | 39.7 |
| 29 | 43.441 | 37.229 | 24 | 1688.0 | 269.5 | 798.2 | 350.8 | 44.0 |
| 30 | 43.143 | 36.906 | 24 | 1768.6 | 280.1 | 788.7 | 319.2 | 40.5 |
| 31 | 43.093 | 37.035 | 24 | 1657.0 | 175.4 | 699.3 | 309.5 | 44.3 |
| 32 | 43.649 | 37.059 | 24 | 1867.0 | 286.8 | 830.1 | 344.3 | 41.5 |
| 33 | 43.893 | 36.741 | 24 | 1425.8 | 274.9 | 644.7 | 246.4 | 38.2 |
| 34 | 43.487 | 37.093 | 24 | 1650.0 | 349.4 | 800.4 | 286.6 | 35.8 |
| 35 | 43.350 | 37.050 | 24 | 2015.0 | 219.2 | 918.0 | 393.4 | 42.9 |
| 36 | 43.269 | 37.115 | 24 | 1677.5 | 316.4 | 774.6 | 316.2 | 40.8 |
| 37 | 43.589 | 36.508 | 24 | 1014.6 | 187.1 | 427.2 | 179.4 | 42.0 |
| 38 | 43.598 | 36.701 | 24 | 1262.5 | 201.8 | 543.9 | 222.2 | 40.8 |
| | | | | *Sulaimaniyah* | | | | |
| 39 | 45.436 | 35.557 | 24 | 1147.5 | 230.2 | 627.7 | 219.5 | 35.0 |
| 40 | 45.140 | 35.589 | 24 | 1209.8 | 201.6 | 652.9 | 246.5 | 37.8 |
| 41 | 45.974 | 35.186 | 24 | 1081.4 | 295.4 | 658.1 | 215.2 | 32.7 |
| 42 | 45.941 | 35.620 | 24 | 1873.4 | 384.0 | 1017.6 | 341.3 | 33.5 |
| 43 | 45.575 | 35.720 | 24 | 1212.5 | 355.4 | 741.4 | 220.8 | 29.8 |
| 44 | 44.953 | 35.954 | 24 | 1058.2 | 224.6 | 599.5 | 217.7 | 36.3 |
| 45 | 45.133 | 36.176 | 24 | 1374.5 | 271.2 | 723.1 | 257.6 | 35.6 |
| 46 | 44.886 | 36.239 | 24 | 1618.4 | 307.4 | 768.6 | 293.4 | 38.2 |
| 47 | 45.853 | 35.344 | 24 | 1159.9 | 265.0 | 575.8 | 217.3 | 37.7 |
| 48 | 45.390 | 35.309 | 24 | 1727.5 | 103.6 | 798.0 | 350.7 | 44.0 |
| 49 | 45.587 | 35.425 | 24 | 1029.7 | 184.3 | 525.0 | 193.4 | 36.8 |
| 50 | 45.705 | 35.385 | 24 | 1275.0 | 205.4 | 558.3 | 239.4 | 42.9 |
| 51 | 46.116 | 35.225 | 24 | 1300.7 | 285.5 | 700.6 | 246.4 | 35.2 |
| 52 | 45.410 | 35.901 | 24 | 1296.6 | 326.2 | 746.6 | 238.9 | 32.0 |
| 53 | 44.787 | 36.210 | 24 | 1338.6 | 218.1 | 592.1 | 249.8 | 42.2 |
| 54 | 45.686 | 35.116 | 24 | 914.3 | 148.9 | 459.8 | 175.7 | 38.2 |
| 55 | 44.833 | 35.533 | 24 | 681.8 | 106.3 | 320.4 | 121.7 | 38.0 |
| 56 | 44.897 | 35.748 | 24 | 805.0 | 125.0 | 418.6 | 156.5 | 37.4 |
| 57 | 45.030 | 36.066 | 24 | 1213.5 | 241.4 | 695.9 | 241.5 | 34.7 |
| 58 | 45.182 | 35.286 | 24 | 1089.0 | 144.4 | 499.1 | 214.5 | 43.0 |
| 59 | 45.509 | 34.823 | 24 | 900.0 | 139.1 | 389.9 | 175.3 | 45.0 |
| 60 | 44.966 | 34.683 | 24 | 868.8 | 134.3 | 293.1 | 166.2 | 56.7 |

*3.5. The Correlation Coefficient*

The significant spatiotemporal variability of precipitation in the KRI indicates and forecasts an increase in drought frequency and duration. The correlation coefficients between precipitation, SPI, MSAVI2 mean and vegetation area, crop area, and crop production from 1998 to 2021 are shown in Table 6 (average of 24 years). The analysis of variance for drought indices indicated statistically significant differences between the studied years at p of 0.01 and p of 0.05. The results demonstrated a substantial positive correlation between MSAVI2 and precipitation (Table 6). There was a statistically significant correlation between remote sensing-derived spectral indices and precipitation.

**Table 6.** Correlation coefficients between spectral indices, meteorological indices crop area, crop yield, and annual average precipitation.

| | Crop Area (km$^2$) | (LD) Area (km$^2$) | MSAVI2 (km$^2$) | SPI | MSAVI2 (Mean) | Precipitation (mm) | Crop Yield (Ton)/Year |
|---|---|---|---|---|---|---|---|
| Crop Area (km$^2$) | 1 | −0.05 | 0.37 | 0.28 | 0.35 | 0.28 | 0.71 ** |
| (LD) Area (km$^2$) | −0.05 | 1 | 0.33 | 0.68 ** | 0.22 | 0.69 ** | 0.05 |
| MSAVI2 Area(km$^2$) | 0.37 | 0.33 | 1 | 0.69 ** | 0.78 ** | 0.68 ** | 0.73 ** |
| SPI | 0.281 | 0.68 ** | 0.69 ** | 1 | 0.53 * | 0.995 ** | 0.42 |
| MSAVI2 (Mean Value) | 0.35 | 0.22 | 0.77 ** | 0.53 * | 1 | 0.51* | 0.61 ** |
| Precipitation (mm) | 0.28 | 0.69 ** | 0.68 ** | 0.995 ** | 0.51 * | 1 | 0.39 |
| Crop Yield (Ton)/Year | 0.71 ** | 0.05 | 0.73 ** | 0.42 | 0.61 ** | 0.39 | 1 |

* Correlation is significant at the 0.05 level (2-tailed). ** Correlation is significant at the 0.01 level (2-tailed).

Table 6 illustrates the relationship between the mean values of vegetation cover based on MSAVI2 characteristics, elevation, latitude, and longitude (precipitation). The graph indicates that as terrain elevation rises, precipitation and elevation increase, but event duration increases. Consequently, mountain regions receive relatively heavy, strong, and long-lasting precipitation. MSAVI2 and elevation are significantly correlated with the event features of the study region. Surface relief greatly influences land characteristics and productivity [60]. The lowland parts receive less precipitation than the mountainous parts. Nevertheless, MSAVI2 measurements are related to precipitation quantity and elevation [61].

**4. Discussion**

LD's surface area has witnessed major expansions and contractions over the years. Drought estimates are crucial and are required to evaluate how the climates of these lakes and their environs have altered [16]. It is similar to the reports of UNESCO [36] and Fadhil [62]. In addition, the studies show that LD experienced severe droughts in 2008 and 2009. These results are equivalent to those of prior research [15,16]. The years 1999 and 2008 experienced the most severe drought conditions, followed by 2009 and 2012. In 2008, the southeast of the study area, comprising three stations, was the most affected region, with this finding also supported by [63,64]. In addition, the western and southern parts of the study area experienced mild drought conditions in 1999. A previous study [7,65] indicated that the SPI was an effective index; for instance, the SPI at station #9 at the Choman site for the hydrological year 2007–2008 was −2.52, while at station #13 at the Bastora site for the same year it was −1.94. The discrepancy indicates that the precipitation at station #9 (Choman site) in 2007–2008 was less than that at station #13 (Bastora site) in the same time period.

These CV results accord with the findings of [33], indicating a considerable climatic gradient from the south's semi-arid climate to the north's semi-wet climate. This also

supports past rainfall patterns and our region's understanding [66]. Elevation is the most influential factor in the regional variation of rainfall. On the other hand, the CV exhibited the reverse tendency [67]. In addition, the KRI's mountainous areas receive an abundance of seasonal precipitation. We discovered that the high seasonal precipitation in mountainous areas is mostly the result of frequent and prolonged rainstorm episodes. However, seasonal precipitation in certain portions of the border area is characterized by low-intensity, short-duration occurrences [64,65,68]. Nearly everywhere to the south-southwest of the KRI, where precipitation and altitude are frequently limited (Figure 1C,D), the MSAVI2 reported much lower values. At all sites, MSAVI2 levels declined concurrently with lower elevations [69–71].

The majority of KRI regions had a severe drought between 1999 and 2008. However, the drought intensity dropped to moderate in 2000, 2009, and 2012, as confirmed by 90% of KRI weather stations. There have been five major droughts in the previous two decades Other than years of severe and moderate drought, the remaining years experienced drought conditions that were near average. The lowest water levels in LD were recorded in 1999, 2000, 2008, and 2009, which is consistent with the decline in SPI values. The MSAVI2 data, on the other hand, indicated droughts in 2000, 2008, 2012, and 2021. Therefore, SPI is a better indicator of drought in the region than MSAVI2, in which the SPI is dependent entirely on precipitation, whereas vegetation cover (MSAVI2) is affected by a more significant number of factors, such as precipitation, temperature, DEM, and soil qualities [72].

## 5. Conclusions

Between 1999 and 2008, most KRI faced a severe drought, but 90% of KRI weather stations indicated that the drought severity decreased to moderate in 2000, 2009, and 2012. According to the findings, the lowest water levels in LD were recorded in 1999, 2000, 2008, and 2009. This study shed light on historical and agricultural drought events' frequency, length, and spatial extent. Based on the study of rainfall data collected in the KRI from 1998 to 2021, the following may be determined: 1. The yearly precipitation is highest in the northern portion of KRI and lowest in the southern part. 2. The yearly rainfall is quite irregular, with a coefficient of variation of 30%. In the south and southwest of the KRI, the precipitation's CV was reported to vary the most spatially by 56.7%. 3. The SPI data indicated that 2007–2008 was the driest hydrological year between 1998 and 2021. 4. The annual precipitation series exhibits a significant correlation coefficient at most stations. The correlations between the SPI series and the area of LD, vegetation cover, crop area, and crop yield were significant and positive. 5. Between 1999 and 2008, spatial patterns of drought frequency based on the SPI revealed substantial increasing trends of drought severity at stations in the northeast, mid-latitude, and southwest parts of the KRI.

According to the spatiotemporal drought map pattern, the top and middle regions of the KRI had moderate droughts in 1999 and 2008. SPI, NDWI, and MSAVI2 all showed identical drought patterns, consistent with the fall in SPI values. The remainder of the region had acute drought conditions. In contrast, the MSAVI2 data suggested droughts in 2000, 2008, 2012, and 2021. SPI relies solely on precipitation, whereas vegetation cover (MSAVI2) is controlled by a greater variety of parameters, including precipitation, temperature, elevation, latitude, and soil quality [72]. The results of the past 24 years indicate that the drought's consequences were more pronounced in the southern and southeastern regions. In addition, these parts are characterized by expansive grain-growing plains, and the absence of methods to mitigate the consequences of frequent droughts has led to the desertification of these regions. Using MSAVI2 and NDWI, the present work seeks to determine the spatiotemporal extent of drought across KRG and evaluates the performance of the indices by comparing the estimations to the meteorological drought indicator SPI.

In general, we may infer that the drought indicators included in this study demonstrated comparable patterns. Between indices for all analyzed meteorological stations, robust coefficients of determination (R2) were determined. However, it is difficult to infer from this study the precise driving mechanism underlying MSAVI2, as global warming,

climate change, temperature fluctuations, and variation in geopotential height may have all had a substantial effect. Continuous observation of rainfall levels and comparisons with current consumption levels can prevent human-caused drought and aid in developing an intense drought management program [59]. The findings give better insight into the importance of remote sensing applications to better understand the agricultural and water situations in data-scarce regions such as the KRI.

**Author Contributions:** Conceptualization, A.M.F.A.-Q., H.A.A.G. and K.M.; Data curation, H.A.A.G., A.M.F.A.-Q., H.A.S.R., S.H.A., K.H. and S.H.Z.; Formal analysis, H.A.A.G. and A.M.F.A.-Q.; Investigation, A.M.F.A.-Q., C.R., K.M., J.P.M., M.R. and H.A.A.G.; Methodology, H.A.A.G., K.H. and A.M.F.A.-Q.; Resources, A.M.F.A.-Q., C.R., K.M., L.H., H.A.S.R., S.H.Z. and H.A.A.G.; Supervision, A.M.F.A.-Q.; Validation, H.A.A.G., S.H.A. and A.M.F.A.-Q.; Visualization, A.M.F.A.-Q., C.R., M.R., K.H. and H.A.A.G.; Writing—original draft, A.M.F.A.-Q. and H.A.A.G.; Writing—review and editing, A.M.F.A.-Q., M.R., C.R., J.P.M. and K.M. All authors have read and agreed to the published version of the manuscript.

**Funding:** This study has received partial funding from Nuffic, the Orange Knowledge Programme, through the OKP-IRA-104278 project titled "Efficient water management in Iraq switching to climate smart agriculture: capacity building and knowledge development", Coordinated by Wageningen University and Research, The Netherlands and Salahaddin University, Erbil, Kurdistan Region, Iraq.

**Institutional Review Board Statement:** Not applicable.

**Informed Consent Statement:** Not applicable.

**Data Availability Statement:** Some data in this manuscript were obtained from the Ministry of Agriculture and Water Resources, Kurdistan Region, Iraq, while other data was provided by the United States Geological Service (USGS), along with Landsat images being freely available on its website and statistical analysis of the parameters.

**Acknowledgments:** The authors would like to thank the United States Geological Service (USGS) for providing the Landsat images freely on its website. The authors would like to thank Tariq H. Kakahama and Fuad M. Ahmad at the College of Agricultural Engineering Sciences, Salahaddin University, for their sincere assistance. We are extremely grateful to the anonymous reviewers for their insightful comments and suggestions that significantly enhanced the quality of our paper. We are also thankful to Nuffic, the Orange Knowledge Programme, through the OKP-IRA-104278, Wageningen University and Research, The Netherlands; the Ministry of Agriculture and Water Resources, Water Resources Department, Salahaddin University; and the Tishk International University, Erbil, Kurdistan Region, Iraq, for their valuable support.

**Conflicts of Interest:** The authors declare no conflict of interest.

## Appendix A

**Table A1.** The Annual Precipitation (AP) (mm) (average of 24 years), DEM, and coordinates (latitude and longitude) of the 60 meteorological stations in the IKR used in this study.

| Station No. | Station Name | Lat- | Long- | DEM (m) | AP (mm) | Station No. | Station Name | Lat- | Long- | DEM (m) | AP (mm) |
|---|---|---|---|---|---|---|---|---|---|---|---|
| 1 | Erbil | 36.1911 | 44.0092 | 412.7 | 337.3 | 31 | Mangish | 37.0351 | 43.0925 | 1030.2 | 689.0 |
| 2 | Qushtapa | 36.0009 | 44.0285 | 390.8 | 301.3 | 32 | Deraluke | 37.0586 | 43.6493 | 706.8 | 819.5 |
| 3 | Khabat | 36.2728 | 43.6739 | 285.9 | 317.0 | 33 | Akre | 36.7414 | 43.8933 | 683.1 | 633.7 |
| 4 | Bnaslawa | 36.1538 | 44.1400 | 540.7 | 338.9 | 34 | Amadia | 37.0925 | 43.4872 | 1148.5 | 790.7 |
| 5 | Harir | 36.5511 | 44.3648 | 837.3 | 576.8 | 35 | Sarsink | 37.0503 | 43.3503 | 957.1 | 905.9 |
| 6 | Soran | 36.6385 | 44.5614 | 701.6 | 647.2 | 36 | Bamarni | 37.1151 | 43.2693 | 1203.0 | 763.4 |
| 7 | Shaqlawa | 43.9851 | 36.2094 | 966.5 | 762.9 | 37 | Bardarash | 36.5082 | 43.5894 | 363.6 | 418.4 |
| 8 | Khalifan | 36.5986 | 44.4038 | 697.1 | 699.3 | 38 | Qasrok | 36.7009 | 43.5980 | 414.8 | 533.7 |
| 9 | Choman | 36.6374 | 44.8893 | 1178.4 | 750.8 | 39 | SU | 35.5572 | 45.4356 | 870.8 | 617.3 |
| 10 | Sidakan | 36.7974 | 44.6714 | 1011.3 | 835.3 | 40 | Bazian | 35.5890 | 45.1395 | 943.7 | 652.9 |

**Table A1.** *Cont.*

| Station No. | Station Name | Lat- | Long- | DEM (m) | AP (mm) | Station No. | Station Name | Lat- | Long- | DEM (m) | AP (mm) |
|---|---|---|---|---|---|---|---|---|---|---|---|
| 11 | Rwanduz | 36.6119 | 44.5247 | 801.6 | 719.6 | 41 | Halabja | 35.1864 | 45.9739 | 716.6 | 641.4 |
| 12 | Mergasur | 36.8382 | 44.3062 | 1108.9 | 1370.3 | 42 | Penjwen | 35.6197 | 45.9414 | 1442.9 | 1004.2 |
| 13 | Dibaga | 35.8730 | 43.8050 | 328.3 | 267.5 | 43 | Chwarta | 35.7197 | 45.5747 | 1011.6 | 741.1 |
| 14 | Gwer | 36.0449 | 43.4808 | 309.7 | 256.6 | 44 | Dukan | 35.9542 | 44.9528 | 700.4 | 586.4 |
| 15 | Barzewa | 36.6268 | 44.6333 | 798.3 | 722.9 | 45 | Qaladiza | 36.1755 | 45.1333 | 628.2 | 711.7 |
| 16 | Bastora | 36.3389 | 44.1605 | 630.0 | 436.8 | 46 | Rania | 36.2391 | 44.8855 | 607.8 | 753.5 |
| 17 | Makhmoor | 35.7833 | 43.5833 | 287.7 | 244.3 | 47 | Said Sadiq | 35.3437 | 45.8534 | 544.1 | 564.6 |
| 18 | Koya | 36.0994 | 44.6481 | 724.5 | 501.8 | 48 | Qaradagh | 35.3093 | 45.3896 | 887.9 | 784.9 |
| 19 | Taqtaq | 35.8874 | 44.5856 | 397.5 | 386.2 | 49 | Arbat | 35.4246 | 45.5868 | 701.6 | 515.2 |
| 20 | Shamamk | 36.0400 | 43.8467 | 310.6 | 297.4 | 50 | KaniPanka | 35.3850 | 45.7046 | 685.8 | 549.6 |
| 21 | Duhok | 36.8679 | 42.9790 | 588.3 | 520.0 | 51 | Byara | 35.2251 | 46.1163 | 1333.5 | 693.3 |
| 22 | Semel | 36.8733 | 42.8540 | 491.6 | 445.2 | 52 | Mawat | 35.9007 | 45.4105 | 1063.8 | 735.4 |
| 23 | Zakho | 37.1436 | 42.6819 | 501.4 | 547.0 | 53 | D-dikhan | 35.1163 | 45.6863 | 534.6 | 577.4 |
| 24 | Batel | 36.9595 | 42.7217 | 531.0 | 461.1 | 54 | Chamchamal | 35.5333 | 44.8333 | 726.6 | 452.5 |
| 25 | Dam-DU | 36.8758 | 43.0029 | 605.6 | 538.3 | 55 | Kalar | 34.6411 | 45.3293 | 243.2 | 313.9 |
| 26 | Dar. hajam | 37.1988 | 42.8227 | 649.8 | 533.7 | 56 | Agjalar | 35.7483 | 44.8974 | 702.3 | 410.6 |
| 27 | Zaxo-farh | 37.1599 | 42.6587 | 447.1 | 542.6 | 57 | Bngrd | 36.0660 | 45.0299 | 841.2 | 683.5 |
| 28 | Batifa | 37.1840 | 37.1840 | 930.2 | 713.6 | 58 | Sangaw | 35.2862 | 45.1825 | 704.4 | 484.9 |
| 29 | Kani Masi | 37.2291 | 37.2291 | 1332.3 | 795.6 | 59 | Bawanor | 34.8233 | 45.5087 | 358.4 | 379.9 |
| 30 | Zaweta | 36.9058 | 36.9058 | 1006.4 | 775.6 | 60 | Kifri | 34.6833 | 44.9664 | 238.7 | 279.2 |

**Table A2.** Landsat data chosen for analysis were a mixture of Landsat TM5, ETM7, and Landsat OLI8.

| Date Years | Sensor | Target_WRS_Path Target_WRS_Row Path/Row | Date_Acquired | Resolutions |
|---|---|---|---|---|
| 1998 | Landsat 5 TM | 170/34,170/35, 169/35, 169/34, 168/35, 168/36 | 10/04, 10/04, 21/05, 21/05, 30/05, 30/05 | 30 m |
| 1999 | Landsat 5 TM | 170/34,170/35, 169/35, 169/34, 168/35, 168/36 | 13/04, 13/04, 22/04, 22/04,01/05, 01/05 | 30 m |
| 2000 | Landsat 5 TM Landsat 7 ETM+ | 170/34, 170/35, 169/35, 169/34, 168/35, 168/36 | 15/05, 15/05, 16/04, 16/04, 25/04, 25/04 | 30 m |
| 2001 | Landsat 7 ETM+ | 170/34,170/35, 169/35, 169/34, 168/35, 168/36 | 26/04, 26/04, 21/05, 21/05, 28/04, 28/04 | 30 m |
| 2002 | Landsat 7 ETM+ | 170/34,170/35, 169/35, 169/34, 168/35, 168/36 | 13/04, 13/04, 08/05, 08/05, 01/05, 01/05 | 30 m |
| 2003 | Landsat 7 ETM+ | 170/34,170/35, 169/35, 169/34, 168/35, 168/36 | 02/05, 02/05, 11/05, 11/05, 20/05, 20/05. | 30 m |
| 2004 | Landsat 7 ETM+ | 170/34,170/35, 169/35, 169/34, 168/35, 168/36 | 06/05, 06/05,11/04, 27/04, 06/05, 06/05 | 30 m |
| 2005 | Landsat 7 ETM+ | 170/34,170/35, 169/35, 169/34, 168/35, 168/36 | 23/04, 23/04, 30/04, 30/04, 23/04, 23/04 | 30 m |
| 2006 | Landsat 7 ETM+ | 170/34,170/35, 169/35, 169/34, 168/35, 168/36 | 26/05, 26/05, 19/05, 19/05, 12/05, 28/05 | 30 m |
| 2007 | Landsat 5 TM Landsat 7 ETM+ | 170/34,170/35, 169/35, 169/34, 168/35, 168/36 | 05/05,05/05, 20/04, 13/04, 07/05, 07/05 | 30 m |
| 2008 | Landsat 7 ETM+ | 170/34,170/35, 169/35, 169/34, 168/35, 168/36 | 15/05, 15/05, 22/04, 24/05, 15/04, 15/04 | 30 m |
| 2009 | Landsat 5 TM Landsat 7 ETM+ | 169/35, 169/34, 170/34,170/35, 168/35, 168/36 | 03/05, 03/05, 02/05, 02/05, 20/05, 20/05 | 30 m |
| 2010 | Landsat 5 TM Landsat 7 ETM+ | 170/34,170/35, 169/35, 169/34, 168/35, 168/36 | 26/05, 29/05, 22/05, 04/04, 05/04, 19/04 | 30 m |
| 2011 | Landsat 5 TM Landsat 7 ETM+ | 170/34,170/35, 169/34, 168/35, 168/36169/35, | 16/05, 16/05, 08/05, 16/04, 16/04, 15/04 | 30 m |
| 2012 | Landsat 7 ETM+ | 170/34,170/35, 169/35, 169/34, 168/35, 168/36 | 26/04, 26/04, 19/05, 19/05, 26/04, 26/04 | 30 m |
| 2013 | Landsat 8 OLI | 170/34,170/35, 169/35, 169/34, 168/35, 168/36 | 05/05, 05/05, 28/04, 28/04, 23/05, 23/05, | 30 m |
| 2014 | Landsat 8 OLI | 170/34,170/35, 169/35, 169/34, 168/35, 168/36 | 06/04, 06/04, 15/04, 01/05, 24/04, 24/04 | 30 m |
| 2015 | Landsat 8 OLI | 170/34,170/35, 169/35, 169/34, 168/35, 168/36 | 09/04, 25/04,18/04, 01/04, 27/04, 27/04 | 30 m |
| 2016 | Landsat 8 OLI | 170/34,170/35, 169/35, 169/34, 168/35, 168/36 | 13/05, 13/05, 20/04, 20/04, 15/05, 15/05 | 30 m |
| 2017 | Landsat 8 OLI | 170/34,170/35, 169/35, 169/34, 168/35, 168/36 | 30/04, 30/04, 09/05, 09/05, 18/05, 18/05 | 30 m |
| 2018 | Landsat 8 OLI | 170/34,170/35, 169/35, 169/34, 168/35, 168/36 | 04,10/04, 10/04, 26/04, 19/04, 19/04 | 30 m |
| 2019 | Landsat 8 OLI | 170/34,170/35, 169/35, 169/34, 168/35, 168/36 | 4/04, 4/04, 13/04, 13/04, 24/05, 24/05 | 30 m |
| 2020 | Landsat 8 OLI | 170/34,170/35, 169/35, 169/34, 168/35, 168/36 | 08/05, 08/05, 15/04, 15/04, 23/03, 23/03 | 30 m |
| 2021 | Landsat 8 OLI | 170/34,170/35, 169/35, 169/34, 168/35, 168/36 | 25/4, 10/05, 20/04, 20/04, 26/03, 26/03 | 30 m |

**Table A3.** The duration, frequency, and severity of droughts based on the SPI index in 60 weather stations in the KRI from 1998 to 2021.

| Station No. | Long- | Lat- | 1997–1998 | 1998–1999 | 1999–2000 | 2000–2001 | 2001–2002 | 2002–2003 | 2003–2004 | 2004–2005 | 2005–2006 | 2006–2007 | 2007–2008 | 2008–2009 |
|---|---|---|---|---|---|---|---|---|---|---|---|---|---|---|
| 1 | 44.009 | 36.191 | −0.68 | −1.94 | −0.76 | −0.15 | 0.61 | 1.31 | 1.17 | 0.71 | 0.69 | 0.45 | −1.16 | −0.46 |
| 2 | 44.028 | 36.001 | −1.11 | −1.44 | −1.24 | 0.08 | 0.68 | 0.79 | 0.66 | 0.49 | 0.11 | 0.63 | −0.57 | −0.37 |
| 3 | 43.674 | 36.273 | 0.05 | −1.01 | −0.84 | 0.32 | 0.25 | 0.74 | 0.65 | 0.25 | 0.56 | −0.01 | −1.52 | −0.9 |
| 4 | 44.14 | 36.154 | −0.64 | −1.62 | −0.6 | −0.08 | 0.16 | 0.98 | 1.06 | 0.49 | 0.53 | 0.48 | −1.38 | −0.95 |
| 5 | 44.365 | 36.551 | 0.26 | −1.44 | −1.02 | −0.72 | 0.69 | 0.71 | 0.77 | 0.32 | 0.38 | 0.58 | −1.44 | −0.57 |
| 6 | 44.561 | 36.638 | −0.32 | −0.75 | −1.57 | −1.01 | 0.76 | 0.9 | 0.79 | 0.34 | 0.8 | 0.64 | −1.37 | −0.52 |
| 7 | 43.985 | 36.209 | 0.35 | −1.58 | −1.25 | −0.39 | 0.64 | 1.05 | 0.8 | 0.47 | 0.54 | 0.81 | −1.71 | −0.65 |
| 8 | 44.404 | 36.599 | −0.27 | −1.68 | −1.68 | 0.07 | 0.87 | 0.8 | 0.54 | 0.01 | 0.64 | 0.53 | −1.01 | −0.45 |
| 9 | 44.889 | 36.637 | −0.17 | −2.09 | −1.29 | −0.59 | 0.65 | 0.27 | 1.03 | 0.04 | 0.31 | 0.56 | −1.13 | −0.39 |
| 10 | 44.671 | 36.797 | 0.6 | −1.27 | −1.24 | −0.49 | 0.67 | 0.45 | 0.82 | 0.34 | 0.86 | 0.32 | −1.68 | −0.73 |
| 11 | 44.525 | 36.612 | 1.01 | −1.3 | −0.53 | 0.24 | −0.06 | 0.25 | 0.99 | 0.42 | 0.87 | 0.94 | −2.03 | −0.81 |
| 12 | 44.306 | 36.838 | −0.91 | −1.94 | −1.86 | 0 | 0.71 | 0.16 | 0.54 | 0.27 | 0.94 | 0.22 | −1.29 | −0.56 |
| 13 | 43.805 | 35.873 | −1.12 | −1.4 | −0.78 | −0.2 | 0.71 | 1.02 | 0.38 | 0.14 | 0.77 | 0.42 | −0.94 | −0.42 |
| 14 | 43.481 | 36.045 | −0.82 | −1.22 | −0.47 | 0.13 | 1.05 | 1.75 | 0.22 | 0.07 | 0.41 | −0.91 | −0.82 | −1.07 |
| 15 | 44.633 | 36.627 | 0.53 | −0.99 | −1.34 | 0.55 | 0.3 | 2.89 | 0.39 | 0.23 | 0.23 | 0.61 | −1.81 | −0.91 |
| 16 | 44.16 | 36.339 | 0.52 | −0.74 | −0.53 | −0.08 | 0.69 | 0.61 | 0.57 | −0.13 | −0.14 | −0.53 | −1.72 | −1.57 |
| 17 | 43.583 | 35.783 | −0.52 | −1.45 | −0.46 | 0 | 0.93 | 1.28 | 0.9 | 0.3 | 0.63 | 0.29 | −0.97 | −0.82 |
| 18 | 44.648 | 36.099 | 0.23 | −1.01 | −0.62 | −0.61 | 0.04 | 0.52 | −0.22 | −0.32 | 0.1 | 0.88 | −1.43 | −1.15 |
| 19 | 44.586 | 35.887 | 0.56 | −0.89 | −1 | −0.41 | 0.04 | 0.35 | 0.47 | 0.18 | 0.27 | 0.51 | −1.72 | −1.56 |
| 20 | 43.847 | 36.04 | −0.53 | −1.62 | −0.45 | 0.04 | 0.81 | 1.56 | 0.8 | −0.17 | 0.21 | 0.17 | −0.84 | −0.68 |
| 21 | 42.979 | 36.868 | −0.11 | −1.26 | −1.39 | 0.4 | 0.24 | 0.9 | 0.31 | 0.3 | 0.77 | 0.11 | −1.4 | −0.92 |
| 22 | 42.854 | 36.873 | 0.08 | −0.99 | −0.63 | 0.65 | 0.19 | 0.47 | 0.55 | 0.19 | 0.62 | 0.4 | −1.83 | −0.96 |
| 23 | 42.682 | 37.144 | 0.58 | −1.58 | −0.74 | 0.14 | 0.49 | 0.74 | 0.25 | 0.17 | 0.53 | 0.32 | −1.71 | −0.9 |
| 24 | 42.722 | 36.959 | 0.71 | −0.9 | −1.02 | 0.3 | 0.25 | 0.73 | 0.4 | 0.48 | 0.87 | 0.33 | −1.89 | −0.53 |
| 25 | 43.003 | 36.876 | −0.08 | −1.47 | −0.46 | −0.28 | 0.23 | 0.72 | 0.46 | 0.2 | 0.69 | 0.53 | −1.43 | −1.01 |
| 26 | 42.823 | 37.199 | 0.04 | −1.32 | −1.43 | 0.26 | 0.66 | 1.02 | 0.61 | −0.67 | 0.18 | −0.63 | −0.96 | −0.5 |
| 27 | 42.659 | 37.16 | 0.06 | −1.17 | −1.38 | −0.1 | 0.28 | 0.5 | 0.48 | 0.26 | 0.44 | 0.27 | −1.61 | −0.96 |
| 28 | 43.013 | 37.184 | −0.26 | −1.57 | −1.6 | −0.45 | 0.3 | 0.7 | 0.23 | 0.32 | 0.75 | 0.53 | −0.91 | −0.65 |
| 29 | 43.441 | 37.229 | −0.61 | −1.28 | −1.34 | −1.01 | 0.4 | 0.16 | 0.24 | 0.38 | 0.59 | 0.55 | −1.18 | −0.15 |
| 30 | 43.143 | 36.906 | −0.38 | −1.53 | −0.28 | 0.07 | 0.25 | 0.49 | 0.37 | −0.03 | 0.76 | 0.18 | −0.91 | −1.13 |
| 31 | 43.093 | 37.035 | −0.3 | −1.87 | −1.08 | −0.24 | 0.2 | 0.54 | 0.3 | 0.01 | 0.73 | 0.33 | −1.11 | −0.58 |
| 32 | 43.649 | 37.059 | −0.71 | −1.47 | −1.44 | 0 | 0.55 | 0.4 | 0.64 | −0.11 | 0.71 | 0.32 | −0.69 | −0.75 |
| 33 | 43.893 | 36.741 | 0.72 | −1.26 | −0.74 | −0.15 | 0.23 | 0.52 | 0.36 | 0.25 | 0.5 | 0.25 | −1.03 | −1.39 |
| 34 | 43.487 | 37.093 | 0.06 | −1.4 | −0.8 | −0.45 | 0.5 | 0.23 | −0.07 | −0.15 | 0.32 | 0.67 | −0.99 | −1.03 |
| 35 | 43.35 | 37.05 | −0.73 | −1.83 | −1.14 | 0.25 | 0.54 | 0.28 | 0.09 | 0.13 | 0.57 | 0.19 | −0.96 | −0.89 |
| 36 | 43.269 | 37.115 | −0.64 | −1.34 | −1.29 | −0.21 | 0.78 | 0.25 | 0.1 | 0.06 | 0.93 | 0.51 | −1.1 | −0.93 |
| 37 | 43.589 | 36.508 | 0.25 | −0.71 | −0.67 | −0.49 | −0.31 | 0.79 | 0.75 | 0.67 | 1.0 | 0.33 | −1.23 | −1.23 |
| 38 | 43.598 | 36.701 | −0.06 | −1.11 | −0.93 | 0.04 | 0.3 | 0.57 | 0.55 | 0.44 | 0.89 | 0.19 | −1.4 | −1.46 |
| 39 | 45.436 | 35.557 | 1.28 | −1.78 | −0.83 | −0.21 | 0.71 | 1.0 | 0.92 | 0.28 | 0.6 | 0.11 | −0.92 | −0.66 |
| 40 | 45.14 | 35.589 | 0.7 | −1.28 | −0.64 | 0.05 | 0.4 | 0.69 | 0.5 | 0.35 | 0.41 | 0.17 | −1.59 | −0.91 |
| 41 | 45.974 | 35.186 | 1.62 | −2.16 | −1.38 | −1.01 | 1.08 | 0.76 | 1.46 | 0.96 | 1.17 | 0.32 | −2.14 | −0.77 |
| 42 | 45.941 | 35.62 | −0.13 | −1.68 | −1.74 | −0.65 | 0.72 | 1.02 | 0.64 | 0.3 | 0.69 | 0.41 | −1.19 | −0.76 |
| 43 | 45.575 | 35.72 | 0.81 | −1.28 | −1.1 | −0.4 | 0.35 | 0.46 | 0.58 | 0.22 | 0.42 | −0.03 | −1.15 | −0.78 |
| 44 | 44.953 | 35.954 | 1.71 | −1.28 | −0.83 | −0.41 | 0.65 | 0.76 | 1.17 | 0.98 | 0.41 | 0.22 | −1.85 | −1.38 |
| 45 | 45.133 | 36.176 | 0.01 | −1.68 | −1.37 | −0.48 | 0.91 | 1.23 | 1.05 | 0.15 | 0.13 | −0.43 | −1.19 | −0.47 |
| 46 | 44.886 | 36.239 | 0.99 | −1.35 | −1.05 | −0.24 | 0.72 | 0.78 | 0.87 | 0.49 | 0.15 | 0.48 | −1.44 | −1.06 |
| 47 | 45.853 | 35.344 | 1.59 | −1.26 | −1.27 | −0.83 | 0.81 | 0.47 | 0.48 | −0.07 | 0.81 | 0.12 | −1.47 | −1.0 |
| 48 | 45.39 | 35.309 | 0.59 | −1.15 | −0.86 | −0.33 | 0.43 | 0.48 | 0.37 | 0.28 | 0.46 | 0.1 | −2.25 | −0.93 |
| 49 | 45.587 | 35.425 | 1.55 | −1.49 | −0.5 | −0.46 | 0.74 | 0.42 | 0.34 | 0.02 | 0.32 | 0.02 | −1.74 | −0.92 |
| 50 | 45.705 | 35.385 | 0.79 | −1.29 | −0.9 | −0.68 | 0.5 | 0.22 | 0.19 | 0.09 | 0.8 | 0.08 | −1.27 | −0.81 |
| 51 | 46.116 | 35.225 | 0.95 | −1.42 | −1.46 | −0.61 | 0.65 | 0.58 | 0.57 | 0.38 | −0.69 | 0.06 | −1.1 | −0.64 |
| 52 | 45.411 | 35.901 | 1.28 | −1.23 | −0.86 | −0.86 | 0.72 | 0.69 | 0.9 | 0.38 | −0.49 | 0.23 | −1.61 | −1.14 |
| 53 | 44.787 | 36.21 | 0.62 | −1.45 | −1.15 | −1.0 | 1.13 | 0.84 | 0.59 | 0.56 | 0.42 | −0.2 | −1.62 | −0.78 |
| 54 | 45.686 | 35.116 | 0.25 | −0.86 | −1.12 | 0.01 | 0.54 | 0.72 | 0.77 | 0.6 | −0.03 | −0.55 | −1.66 | −0.88 |
| 55 | 44.833 | 35.533 | 0.78 | −0.06 | 0.1 | 0.16 | 0.96 | −0.16 | −0.17 | 0.2 | −0.03 | −0.53 | −2.09 | −0.73 |
| 56 | 44.897 | 35.748 | 0.45 | −0.73 | −0.92 | −0.28 | 0.6 | 0.99 | 1.01 | 0.76 | 0.45 | −0.22 | −1.83 | −1.09 |
| 57 | 45.03 | 36.066 | 1.29 | −1.24 | −1.04 | −0.29 | 0.86 | 0.64 | 0.94 | 0.84 | 0.43 | 0.27 | −1.98 | −1.02 |
| 58 | 45.183 | 35.286 | 0.62 | −0.81 | −0.85 | −0.28 | 0.61 | 0.48 | 0.57 | 0.18 | 1.24 | 1.09 | −1.97 | −1.12 |
| 59 | 45.509 | 34.823 | 0.66 | −0.48 | −0.54 | 0.35 | 0.7 | 0.22 | −0.04 | 0.2 | −0.67 | −0.51 | −1.67 | −1.07 |
| 60 | 44.966 | 34.683 | 0.91 | −0.68 | −0.56 | 0.15 | 0.17 | −0.75 | −1.15 | −0.38 | −0.1 | −0.22 | −0.56 | −0.19 |

**Table A3.** *Cont.*

| Station No. | Long- | Lat- | 2009–2010 | 2010–2011 | 2011–2012 | 2012–2013 | 2013–2014 | 2014–2015 | 2015–2016 | 2016–2017 | 2017–2018 | 2018–2019 | 2019–2020 | 2020–2021 |
|---|---|---|---|---|---|---|---|---|---|---|---|---|---|---|
| 1 | 44.009 | 36.191 | 0.25 | −0.09 | −1.05 | 0.65 | −0.35 | 0.02 | 0.56 | −0.28 | 0.25 | 1.8 | 0.59 | −1.3 |
| 2 | 44.028 | 36.001 | 0.22 | −0.73 | −1.12 | 0.62 | 0.12 | 0.4 | 0.85 | 0.06 | 0.38 | 1.9 | 1.03 | −1.06 |
| 3 | 43.674 | 36.273 | −0.3 | −0.03 | −0.5 | 0.82 | −0.1 | 0.36 | 0.64 | −0.38 | 0.03 | 2.19 | 1.0 | −0.76 |
| 4 | 44.14 | 36.154 | −0.2 | −0.25 | −0.51 | 0.88 | 0.02 | 0.56 | 0.64 | −0.31 | 0.46 | 1.79 | 0.64 | −0.66 |
| 5 | 44.365 | 36.551 | 0.3 | −0.4 | −0.43 | 0.8 | −0.64 | 0.5 | 0.88 | −0.19 | 0.29 | 1.65 | 0.76 | −0.65 |
| 6 | 44.561 | 36.638 | −0.01 | −0.56 | −0.48 | 0.53 | −0.54 | 0.79 | 0.95 | 0.09 | 0.39 | 1.17 | 0.69 | −0.32 |
| 7 | 43.985 | 36.209 | 0.31 | −0.21 | −0.85 | 1.42 | −0.34 | 0.07 | 0.77 | −0.56 | −0.01 | 1.71 | 0.4 | −1.24 |
| 8 | 44.404 | 36.599 | 0.23 | −0.57 | −0.61 | 0.92 | −0.22 | 0.37 | 0.95 | −0.01 | 0.46 | 1.54 | 0.5 | −0.52 |
| 9 | 44.889 | 36.637 | 0.08 | 0.26 | −0.27 | 1.09 | −0.37 | 0.79 | 1.2 | −0.38 | 0.45 | 1.25 | 0.43 | −0.61 |
| 10 | 44.671 | 36.797 | −0.12 | 0.01 | −0.31 | 0.57 | −1.18 | 0.31 | 1.29 | 0.44 | 0.39 | 1.3 | 0.2 | −0.68 |
| 11 | 44.525 | 36.612 | −0.27 | −0.5 | −0.99 | 0.97 | −0.93 | 0.55 | 1.26 | −0.27 | 0.17 | 1.27 | 0.4 | −1.08 |
| 12 | 44.306 | 36.838 | 0.59 | 0.21 | 0.05 | 1.46 | −0.32 | 0.27 | 1.33 | −0.04 | −0.25 | 1.49 | 0.23 | −0.7 |
| 13 | 43.805 | 35.873 | 0.03 | −0.5 | −0.8 | 0.98 | 0.0 | 0.28 | 0.41 | 0.03 | 0.21 | 2.09 | 1.05 | −0.8 |
| 14 | 43.481 | 36.045 | −0.21 | 0.08 | −0.78 | 0.37 | 0.41 | 0.06 | 0.56 | 0.18 | 0.2 | 1.82 | 1.07 | −0.38 |
| 15 | 44.633 | 36.627 | −0.41 | −0.36 | −0.8 | 0.47 | −0.73 | 0.72 | 0.72 | −0.4 | 0.16 | 0.9 | 0.24 | −0.84 |
| 16 | 44.16 | 36.339 | 0.02 | −0.3 | −0.38 | 1.06 | 0.17 | 0.85 | 0.98 | −0.11 | 0.64 | 1.7 | 0.68 | −0.79 |
| 17 | 43.583 | 35.783 | −0.26 | −0.15 | −1.05 | 0.6 | −0.23 | −0.03 | 0.32 | −0.24 | 0.13 | 1.89 | 1.14 | −0.8 |
| 18 | 44.648 | 36.099 | 0.76 | 0.09 | 0.05 | 0.53 | −0.09 | 0.41 | 1.18 | −0.23 | 0.5 | 1.95 | 0.82 | −0.94 |
| 19 | 44.586 | 35.887 | 0.51 | 0.01 | −0.33 | 0.72 | 0.14 | 0.47 | 1.26 | −0.16 | 0.65 | 1.6 | 0.8 | −1.35 |
| 20 | 43.847 | 36.04 | 0.12 | −0.28 | −1.17 | 0.36 | −0.22 | 0.21 | 0.68 | 0.08 | 0.36 | 2.16 | 0.76 | −1.06 |
| 21 | 42.979 | 36.868 | 0.43 | −0.12 | −1.03 | 1.21 | 0.68 | 0.27 | 0.39 | −0.44 | −0.04 | 1.96 | 0.87 | −1.06 |
| 22 | 42.854 | 36.873 | 0.39 | −0.1.0 | −1.17 | 0.77 | 0.38 | 0.36 | 0.2 | −0.35 | −0.04 | 2.09 | 1.13 | −1.23 |
| 23 | 42.682 | 37.144 | 0.44 | 0.35 | −0.88 | 0.56 | −0.38 | 0.42 | 1.18 | −0.52 | −0.3 | 2.26 | 0.51 | −1.22 |
| 24 | 42.722 | 36.959 | 0.42 | −0.23 | −1.36 | 0.36 | −0.11 | 0.11 | 0.53 | −0.11 | 0.2 | 2.15 | 0.7 | −1.45 |
| 25 | 43.003 | 36.876 | 0.6 | −0.08 | −1.07 | 1.28 | 0.63 | 0.19 | 0.25 | −0.57 | −0.03 | 2.03 | 0.88 | −1.22 |
| 26 | 42.823 | 37.199 | 0.48 | 0.71 | −0.2 | 0.99 | 0.68 | −0.17 | 0.17 | −0.65 | −0.94 | 2.23 | 0.91 | −0.57 |
| 27 | 42.659 | 37.16 | 0.45 | 0.41 | −0.79 | 0.44 | 0.48 | 0.46 | 1.47 | −0.52 | −0.34 | 2.33 | 0.5 | −1.36 |
| 28 | 43.013 | 37.184 | 0.51 | 0.14 | −0.68 | 0.63 | 0.15 | 0.33 | 0.58 | 0.14 | 0.02 | 2.23 | 0.64 | −0.68 |
| 29 | 43.441 | 37.229 | 0.68 | 0.27 | −1.0 | 1.26 | 0.08 | 0.29 | 0.84 | −0.05 | 0.52 | 1.64 | 0.58 | 0.07 |
| 30 | 43.143 | 36.906 | 0.43 | −0.14 | −1.21 | 1.14 | 0.29 | 0.47 | 0.58 | −0.28 | 0.03 | 2.04 | 1.05 | −0.72 |
| 31 | 43.093 | 37.035 | 0.61 | 0.06 | −1.02 | 1.02 | 0.43 | 0.9 | 0.47 | −0.17 | −0.23 | 2.06 | 0.87 | −0.54 |
| 32 | 43.649 | 37.059 | 0.28 | 0.47 | −0.75 | 0.9 | −0.15 | 0.23 | 0.82 | −0.18 | 0.26 | 1.94 | 0.9 | −0.44 |
| 33 | 43.893 | 36.741 | 0.7 | 0.44 | −1.26 | 1.01 | 0.13 | 0.12 | 0.44 | −0.59 | 0.04 | 2.13 | 0.67 | −0.84 |
| 34 | 43.487 | 37.093 | 0.5 | 0.3 | −0.68 | 1.28 | −0.01 | 0.46 | 0.76 | −0.34 | 0.18 | 1.97 | 0.79 | −0.53 |
| 35 | 43.35 | 37.05 | 0.37 | 0.15 | −0.71 | 1.2 | 0.15 | 0.53 | 0.93 | 0.03 | 0.45 | 1.8 | 0.74 | −0.42 |
| 36 | 43.269 | 37.115 | 0.66 | 0.4 | −0.97 | 1.02 | 0.13 | 0.32 | 0.92 | −0.33 | 0.19 | 1.95 | 0.55 | −0.59 |
| 37 | 43.589 | 36.508 | 0.22 | 0.44 | −1.2 | 0.57 | −0.38 | 0.38 | 0.3 | −0.38 | 0.17 | 2.22 | 0.79 | −1.0 |
| 38 | 43.598 | 36.701 | 0.47 | 0.46 | −0.9 | 0.71 | 0.09 | 0.47 | 0.37 | −0.52 | 0.22 | 2.12 | 0.7 | −0.86 |
| 39 | 45.436 | 35.557 | 0.76 | −0.04 | −0.12 | −0.62 | −0.58 | −1.01 | 0.65 | −0.1 | 0.24 | 1.72 | 0.59 | −0.88 |
| 40 | 45.14 | 35.589 | 0.58 | −0.32 | −0.5 | −0.23 | −0.06 | 0.07 | 0.69 | −0.14 | 0.38 | 1.33 | 1.42 | 0.19 |
| 41 | 45.974 | 35.186 | 1.2 | 0.03 | −0.16 | 0.26 | −0.78 | −0.37 | 0.84 | −0.65 | −0.48 | 1.98 | −0.43 | −2.51 |
| 42 | 45.941 | 35.62 | 0.65 | −0.03 | 0.17 | 0.26 | −0.07 | −0.07 | 1.02 | −0.09 | 0.42 | 1.73 | 0.2 | −0.64 |
| 43 | 45.575 | 35.72 | 0.75 | −0.08 | −0.47 | −0.06 | −0.08 | 0.23 | 0.68 | 0.01 | 0.44 | 1.32 | 1.07 | 0.13 |
| 44 | 44.953 | 35.954 | 0.05 | −0.35 | −0.61 | 0 | −0.47 | 0.17 | 0.94 | −0.15 | 0.42 | 1.66 | 0.03 | −1.39 |
| 45 | 45.133 | 36.176 | 0.34 | 0.03 | −0.09 | 0.47 | 0.11 | −0.04 | 1.01 | −0.31 | 0.59 | 1.78 | 0.29 | −0.79 |
| 46 | 44.886 | 36.239 | 0.41 | −0.09 | −0.49 | 0.37 | −0.35 | −0.06 | 0.64 | −0.48 | 0.41 | 1.96 | 0.61 | −0.99 |
| 47 | 45.853 | 35.344 | 0.7 | −0.03 | −0.46 | 0.08 | −0.24 | −0.04 | 1.3 | −0.03 | 0.15 | 2.1 | −0.09 | −1.19 |
| 48 | 45.39 | 35.309 | 0.4 | −0.19 | −0.14 | 0.02 | 0.24 | 0.1 | 1.32 | 0.25 | 0.66 | 1.67 | 0.88 | −0.51 |
| 49 | 45.587 | 35.425 | 0.74 | 0.03 | −0.39 | 0.09 | −0.1 | −0.04 | 0.97 | −0.24 | 0.42 | 1.84 | 0.54 | −0.99 |
| 50 | 45.705 | 35.385 | 0.76 | 0.13 | −0.39 | 0.06 | −0.08 | −0.08 | 0.82 | −0.07 | 0.85 | 1.9 | 0.91 | −0.54 |
| 51 | 46.116 | 35.225 | 0.8 | −0.02 | −0.15 | 0.23 | −0.04 | 0.12 | 0.99 | 0.06 | 0.19 | 1.63 | 0.93 | −0.4 |
| 52 | 45.411 | 35.901 | 0.69 | −0.11 | −0.13 | 0.26 | −0.34 | 0.19 | 0.9 | 0.01 | 0.43 | 1.66 | 0.37 | −0.84 |
| 53 | 44.787 | 36.21 | 0.86 | 0.3 | −0.53 | 0.35 | −0.22 | −0.15 | 1.22 | −0.25 | 0.1 | 2.23 | 0.1 | −1.48 |
| 54 | 45.686 | 35.116 | 0.5 | −0.25 | −1.2 | 0.56 | 0.37 | 0.29 | 1.12 | 0.07 | 0.52 | 1.72 | 0.69 | −0.66 |
| 55 | 44.833 | 35.533 | 0.66 | −0.43 | −1.46 | 0.62 | 0.34 | −0.1 | 1.68 | −0.21 | −0.18 | 2.29 | 0.53 | −1.91 |
| 56 | 44.897 | 35.748 | 0.49 | −0.08 | −1.11 | 0.04 | −0.13 | 0.19 | 0.86 | −0.11 | 0.53 | 1.69 | 0.71 | −0.93 |
| 57 | 45.03 | 36.066 | 0.64 | −0.18 | −0.47 | −0.03 | −0.24 | −0.25 | 0.96 | −0.51 | 0.41 | 1.65 | 0.34 | −1.29 |
| 58 | 45.183 | 35.286 | 0.88 | −0.78 | −1.12 | −0.1 | 0.06 | 0.01 | 1.15 | −0.26 | 0.12 | 2.13 | 0.12 | −1.64 |
| 59 | 45.509 | 34.823 | 0.8 | −0.06 | −1.04 | 0.51 | 0.33 | 0.02 | 1.98 | −0.32 | 0.22 | 2.29 | −0.07 | −1.08 |
| 60 | 44.966 | 34.683 | 1.24 | −0.31 | −1.05 | −0.04 | 0.65 | −0.28 | 2.82 | −0.23 | −0.04 | 1.85 | 0.47 | −1.39 |

**Table A4.** Google Earth Engine JavaScript for estimating MSAVI2.

```
/** Kawa Hakzi 2022 kawahakzy@gmail.com MSAVI2 */
// Assign a common name to the sensor-specific bands.
var LC9_BANDS = ['B2', 'B3', 'B4', 'B5', 'B6', 'B7', 'B10']; //Landsat 8
var LC8_BANDS = ['B2', 'B3', 'B4', 'B5', 'B6', 'B7', 'B10']; //Landsat 8
var LC7_BANDS = ['B1', 'B2', 'B3', 'B4', 'B5', 'B7', 'B6_VCID_2']; //Landsat 7
var LC5_BANDS = ['B1', 'B2', 'B3', 'B4', 'B5', 'B7', 'B6']; //Llandsat 5
var STD_NAMES = ['blue', 'green', 'red', 'nir', 'swir1', 'swir2', 'temp'];
var l9 = ee.ImageCollection('LANDSAT/LC09/C02/T1_TOA').select(LC9_BANDS,
STD_NAMES)// Landsat 8
//Bands are not arranged yet
var l8 = ee.ImageCollection('LANDSAT/LC08/C01/T1_TOA').select(LC8_BANDS,
STD_NAMES)// Landsat 8
//print(l8, 'Landsat 8')
var l7 = ee.ImageCollection('LANDSAT/LE07/C01/T1_TOA').select(LC7_BANDS, STD_NAMES)
//Landsat 7
//print(l7, 'Landsat 7')
var l5 = ee.ImageCollection('LANDSAT/LT05/C01/T1_TOA').select(LC5_BANDS, STD_NAMES)
//Landsat 5
//print(l5, 'Landsat 5')
var images = ee.ImageCollection(l5.merge(l7).merge(l8));//.merge(l9)
var table = ee.FeatureCollection("projects/ee-kawa/assets/kurdistan"),
Map.addLayer(table);
//var images = ee.ImageCollection('LANDSAT/LC08/C01/T1_TOA')
.filterBounds(table)
.filterDate('2019-04-01', '2019-05-01')
.select('B4', 'B5', 'B2', 'B3');
print(images.size());
var nir = images.select('B5');
var red = images.select('B4');
var ndvi = nir;
var clipnir = nir.filterBounds(table).mosaic().clip(table);
var clipred = red.filterBounds(table).mosaic().clip(table);
var msavi2imgmosaic = clipnir.multiply(2).add(1)
.subtract(clipnir.multiply(2).add(1).pow(2)
.subtract(clipnir.subtract(clipred).multiply(8)).sqrt()
).divide(2).rename("MSAVI2");
Map.addLayer(msavi2imgmosaic);
Map.centerObject(table, 7);
Export.image.toDrive({
image: msavi2imgmosaic,
description: 'imageToDrive_year()',
crs: 'EPSG:4326',
scale: 30,
maxPixels:200000000,
region: table )};
```

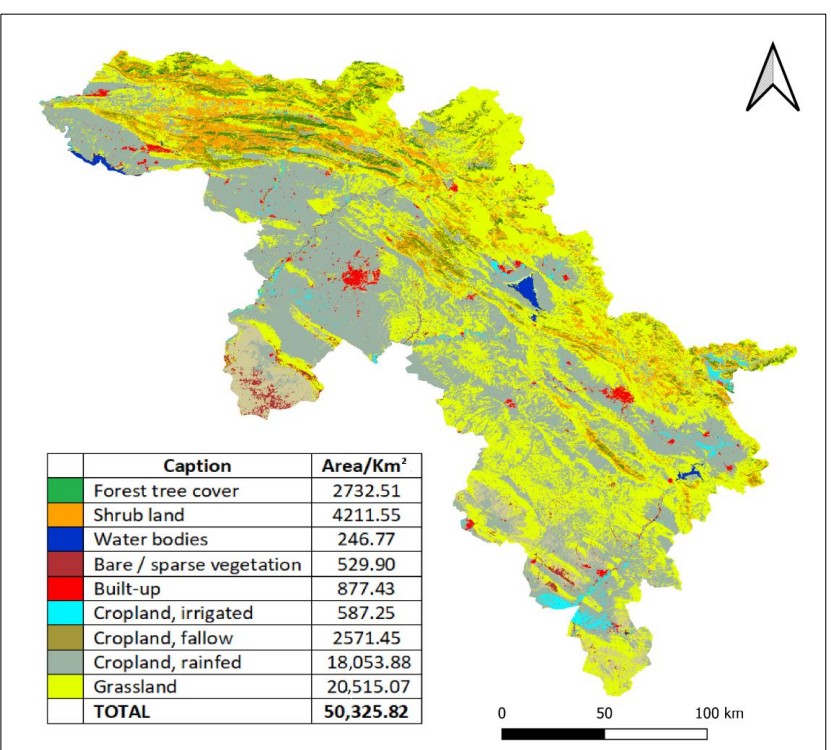

**Figure A1.** Land use and land cover classes in the KRI.

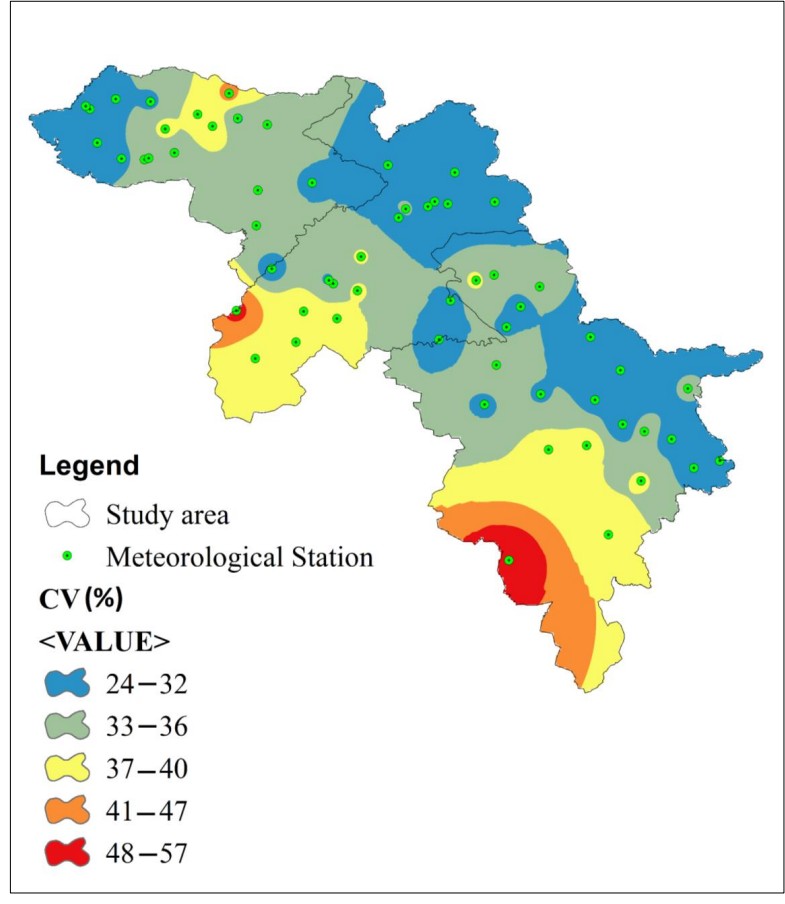

**Figure A2.** The coefficient of variation (CV%) map of annual precipitation for 60 selected meteorological stations across the KRI.

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
