# Peer review of "Drought Severity and Frequency Analysis Aided by Spectral and Meteorological Indices in the Kurdistan Region of Iraq"

_water, doi:10.3390/w14193024_

Round 1

Reviewer 1 Report

see the attached file.

Author Response

Dear Reviewer,

We are very grateful for your valuable suggestions, which have given us quite valuable insight on how to further improve our manuscript for the acceptance for publication in the esteemed journal (Journal of Water).

Kindly, we have revised the manuscript. All the revised sentences, figures, tables, words, and numbers are colored by a yellow background. The revisions were done based on your precious comments and suggestions.

We are now pleased to submit the revised manuscript along with the detailed responses below.

Your reconsiderations are highly appreciated.

Thank you very much again.

Yours sincerely,

Heman A. Gaznayee

Reviewer 2 Report

The authors propose an extensive and interesting study based on the spatio-temporal analysis of drought characteristics in the Iraqi Kurdistan region, using a combination of well-known methodologies.

In my opinion, it is a good article, however, it can be improved on some aspects.

Firstly, I feel that the manuscript is excessively long, which leads to the reader's attention waning before the conclusion. Therefore, I think the article can be shortened without compromising its content. In particular, the section on the methodology can be shortened, as it contains information that is widely known in the literature.

Next, I believe that the authors should highlight the innovative elements of the study that are not solely related to the specific case study. This is a very important aspect.

It is also important to point out the limitations of this study.

In addition to reducing the length of the paper, it is important to pay more attention to the style of writing, to facilitate comprehension and the importance of the results.

Furthermore, the state of the art must be improved. There are some recent relevant papers on the characterization of drought on the basis of Spectral and Meteorological Indices that have not been considered.

Finally, there are numerous typos in the text.

Author Response

(The authors gave the same response as above.)

Round 2

Reviewer 2 Report

The authors have only minimally considered my comments, and the article has not improved on the initially submitted version.

I had mainly asked for the article to be shortened, in particular by summarising the section on methodology, which does not present a significant novelty, to better highlight the originality of the research, and to consider additional recent bibliographical references on the topic.

None of this was done. I would invite the authors to put more effort into improving the manuscript if the Editor agrees. 

Author Response

Dear Editor and Reviewer

Thank you very much for your email and your concern, which we very much appreciate. Also, we are extremely appreciative of your and the reviewers' insightful comments and suggestions, which have provided us with invaluable insight into how to enhance our paper to meet the standards of the Water journal published by MDPI.

The manuscript has undergone a second round of revisions. The track changes mode displays all the revised texts, figures, tables, words, and numbers. Based on the insightful comments and ideas, revisions were made.

Therefore, we are glad to submit the revised version of our manuscript and comprehensive replies. We highly appreciate your reconsideration.

Best regards,

Thank you very much again.

Yours sincerely,

Heman A. Gaznayee

Round 3

Reviewer 2 Report

I have no further comments.